# PHF13 is a molecular reader and transcriptional co-regulator of H3K4me2/3

Ho-Ryun Chung[1†], Chao Xu[2†‡], Alisa Fuchs[1†], Andreas Mund[3§], Martin Lange[4¶], Hannah Staege[3], Tobias Schubert[3**], Chuanbing Bian[2], Ilona Dunkel[1], Anton Eberharter[5], Catherine Regnard[5], Henrike Klinker[5], David Meierhofer[1], Luca Cozzuto[4,6], Andreas Winterpacht[7], Luciano Di Croce[4,6,8], Jinrong Min[2], Hans Will[3], Sarah Kinkley[1,3*]

[1]Max Planck Institute for Molecular Genetics, Berlin, Germany; [2]Structural Genomics Consortium, Toronto, Canada; [3]Heinrich-Pette-Institute - Leibniz Institute for Experimental Virology, Hamburg, Germany; [4]Centre for Genomic Regulation, Barcelona, Spain; [5]Adolf-Butenandt-Institute and Center for Integrated Protein Science, Ludwig-Maximilians-University, Munich, Germany; [6]Universitat Pompeu Fabra (UPF), Barcelona, Spain; [7]Human Genetics, Friedrich-Alexander-Universität Erlangen-Nürnberg, Erlangen, Germany; [8]Institució Catalana de Recerca i Estudis Avançats, Pg. Lluis Companys, Barcelona, Spain

*For correspondence: kinkley@molgen.mpg.de

[†]These authors contributed equally to this work

Present address: [‡]School of Life Sciences, University of Science and Technology of China, Hefei, Anhui, People's Republic of China; [§]Chromatin Structure and Function Group, The Novo Nordisk Foundation Center for Protein Research, Faculty of Health and Medical Sciences, University of Copenhagen, Copenhagen, Denmark; [¶]TRG-ONCI, Bayer Pharma AG, Berlin, Germany; [**]Perutz Laboratories, Medical University of Vienna, Vienna, Austria

Competing interests: The authors declare that no competing interests exist.

**Abstract** PHF13 is a chromatin affiliated protein with a functional role in differentiation, cell division, DNA damage response and higher chromatin order. To gain insight into PHF13's ability to modulate these processes, we elucidate the mechanisms targeting PHF13 to chromatin, its genome wide localization and its molecular chromatin context. Size exclusion chromatography, mass spectrometry, X-ray crystallography and ChIP sequencing demonstrate that PHF13 binds chromatin in a multivalent fashion via direct interactions with H3K4me2/3 and DNA, and indirectly via interactions with PRC2 and RNA PolII. Furthermore, PHF13 depletion disrupted the interactions between PRC2, RNA PolII S5P, H3K4me3 and H3K27me3 and resulted in the up and down regulation of genes functionally enriched in transcriptional regulation, DNA binding, cell cycle, differentiation and chromatin organization. Together our findings argue that PHF13 is an H3K4me2/3 molecular reader and transcriptional co-regulator, affording it the ability to impact different chromatin processes.

## Introduction

PHF13, also known as **S**urvival time associated **PH**D finger in **O**varian **C**ancer 1 (SPOC1), is a chromatin affiliated protein that is conserved from zebra fish to humans. PHF13 has been shown to modulate various processes including development (*Bördlein et al., 2011*), DNA damage (*Frohns et al., 2014*; *Mund et al., 2012*), cell cycle (*Kinkley et al., 2009*), antiviral host cell response (*Schreiner et al., 2013*) and higher order chromatin structure (*Kinkley et al., 2009*; *Mund et al., 2012*), underlining its biological importance. Furthermore, PHF13 expression and chromatin localization are temporally regulated (*Kinkley et al., 2009*) and its misregulation correlates with malignant phenotypes (*Mohrmann et al., 2005*) and defective differentiation (*Bördlein et al., 2011*). Together these observations argue that altered PHF13 expression has consequential outcomes and further underscores the necessity of understanding the molecular interplay and contexts governing PHF13 function.

**eLife digest** In human and other eukaryotic cells, DNA is packaged around proteins called histones to form a structure known as chromatin. Chemical tags added to the histones alter how the DNA is packaged and the activity of the genes encoded by that DNA. For example, many active genes are packaged around histone H3 proteins that have "Lysine 4 tri-methyl" tags attached to them. Another protein that is associated with chromatin is called PHF13 and it has several roles, including repairing damaged DNA. However, it was not known whether PHF13 binds to chromatin via the chemical tags, or in another way.

Ho-Ryun, Xu, Fuchs et al. used several biochemical techniques in mouse and human cells to explore how PHF13 specifically interacts with chromatin. These experiments showed that PHF13 binds specifically to DNA and to two types of methyl tags (lysine 4-tri-methyl or lysine 4-di-methyl). These chemical tags are predominantly found at active promoters as well as at a small subset of less active promoters known as bivalent promoters. PHF13 interacted with other proteins on the chromatin that are known to either drive or repress gene activity and it's depletion affected the activity of many genes. Whether PHF13 increased or decreased gene activity depended on whether it was bound to active or bivalent promoters. The active promoters targeted by PHF13 had higher numbers of the tri-methyl tags whereas the di-methyl tags were more common on the bivalent promoters.

These findings provide preliminary evidence that a protein binding to different methyl tags in the same place on histone H3 can have opposite effects on gene activity. Ho-Ryun, Xu, Fuchs et al. now intend to find out more about the other proteins that interact with PHF13 on chromatin.

To date many chromatin binding domains recognizing specific histone posttranslational modifications, have been identified. Included in these are PHD domains which are recruited predominantly to methylated lysine residues (*He et al., 2013*; *Li et al., 2006*; *Mansfield et al., 2011*; *Peña et al., 2006*; *Shi et al., 2006*; *Wysocka et al., 2006*; *Xie et al., 2012*) with a few exceptions (*Ali et al., 2012*; *Hu et al., 2009*; *Lan et al., 2007*; *Lange et al., 2008*; *Mansfield et al., 2011*; *Org et al., 2008*; *Tsai et al., 2010*). The ability of different PHD domain containing proteins to recognize the same modified residue argues that additional factors may regulate their recruitment. Aside from differences in binding affinities, the sensitivity to adjacent sequence modifications also contributes to their specificity (*Ali et al., 2013*; *Fiedler et al., 2008*; *Gatchalian et al., 2013*; *Iberg et al., 2008*; *Ramon-Maiques et al., 2007*; *Vermeulen et al., 2007*; *Yuan et al., 2012*) indicating that many PHD domains read the local combinatorial chromatin environment.

While the molecular readers containing PHD domains interact selectively with chromatin, they do so albeit with relatively weak affinities (*Musselman and Kutateladze, 2011*). To achieve a more stable chromatin association, many chromatin readers contact chromatin in a multivalent fashion, via multiple chromatin binding modules (*Adams-Cioaba et al., 2012*; *Lange et al., 2008*; *Liu et al., 2013*; *Patel et al., 2013*; *Rothbart et al., 2013*; *Ruthenburg et al., 2011*) or in complex with other chromatin readers, which cooperatively read multivalent chromatin signatures (*Ballare et al., 2012*; *Nayak et al., 2011*). The majority of characterized PHD domain proteins are known to be affiliated with chromatin modulating complexes (*Morra et al., 2012*; *Todd and Picketts, 2012*; *Wysocka et al., 2006*) and co-regulate important chromatin processes, including epigenetic programming (*Lan et al., 2007*; *Wen et al., 2010*), transcription (*Fortschegger and Shiekhattar, 2011*), DNA repair (*Li et al., 2013*; *Mund et al., 2012*), differentiation (*Bördlein et al., 2011*; *Gatchalian et al., 2013*), cell cycle (*Kinkley et al., 2009*; *Lim et al., 2013*) and higher chromatin order (*Papait et al., 2008*). Underscoring their importance in the co-regulation of chromatin function, the dysregulation or aberrant fusion of several PHD domain containing proteins has been shown to lead to genomic instability and cancer (*Wang et al., 2009*).

PHF13 contains a single C-terminal PHD domain, which we demonstrate biochemically and structurally, is a molecular reader of H3K4me2/3. Additionally, we show that PHF13 directly interacts with DNA via a centrally located domain, indicating that it can form multivalent chromatin interactions. These interactions were confirmed by peptide binding assays, gel shift assays, ChIP sequencing and

x-ray crystallography allowing us to additionally map these interactions and identify key cis-acting molecular determinants that affect PHF13's affinity for chromatin. Furthemore, we utilized mass spectrometry, size exclusion chromatography and co-immunoprecipitation experiments to identify Polycomb repressive complex 2 (PRC2) and RNA polymerase II (RNA PolII) complexes as novel PHF13 chromatin interaction partners. Consistently, PHF13 ChIP sequencing targets co-occurred with CpG rich DNA, H3K4me2/3, PRC2 and the hypophosphorylated, serine 5 and serine 7 phosphorylated forms of RNA PolII in murine embryonic stem cells (mESCs). PHF13 depletion in mESCs resulted in the reduced binding of SUZ12 and RNA PolII S5P to H3K4me3 and H3K27me3 and altered gene expression of a fraction of PHF13 bound genes. Genes that were up regulated upon PHF13 knockdown were enriched in H3K4me2/3, Polycomb and RNA PolII S5P, while down regulated genes were enriched in H3K4me2/3 and RNA PolII S2P and S5P. Finally, PHF13 target genes were enriched in the functional categories of transcription regulation, cell cycle, chromosome organization and differentiation, consistent with earlier publications describing a role of PHF13 in these processes (*Bördlein et al., 2011*; *Kinkley et al., 2009*; *Mohrmann et al., 2005*; *Mund et al., 2012*; *Schreiner et al., 2013*). Together, these findings argue that PHF13 is a transcriptional co-regulator and a novel H3K4me2/3 molecular reader.

## Results

### PHF13 interacts with nucleosomes and DNA

We have previously demonstrated via differential nuclear fractionation experiments that PHF13 is predominantly affiliated with the chromatin fraction of nuclear lysate, implicating a role in chromatin function (*Kinkley et al., 2009*). Therefore, to gain clearer insight into whether PHF13 contacts chromatin directly, we explored the ability of GST-PHF13 and different GST-PHF13 deletion mutants to interact with recombinant mono-nucleosomes (*Figure 1A–C*). Mono-nucleosomes were generated using recombinant histone octamers and either a 200 bp DNA fragment (*Figure 1B*) or a 151 bp DNA fragment (*Figure 1C*) to recapitulate mono-nucleosomes with or without linker DNA. GST-PHF13 was found to very efficiently shift reconstituted mono-nucleosomes with DNA overhangs, similar to ISWI (positive control) and in contrast to GST (*Figure 1B*). We also noted that the free DNA in the reaction was notably absent in GST-PHF13 lanes in comparison to ISWI and GST, suggesting that PHF13 may also interact with DNA. To further map PHF13's interaction with mono-nucleosomes we analyzed the ability of different PHF13 fragments to shift mono-nucleosomes devoid of DNA overhangs (*Figure 1C*). Surprisingly, the PHD domain of PHF13 was not found to interact with the recombinant mono-nucleosomes and the interaction was mapped to the middle region of PHF13 (101–200; *Figure 1C*). These observations indicate that linker DNA is not necessary for PHF13 to affiliate with the nucleosomes and that the middle 100 aa of PHF13 is capable of forming a direct contact with recombinant mono-nucleosomes. Again we noted the lack of free DNA in the PHF13 101–200 lanes suggesting that this region may interact with free and nucleosomal complexed DNA. To test this idea and if a direct interaction with DNA exists, we performed DNA electrophoretic mobility shift assays (EMSA) using full-length GST-PHF13 and GST-PHF13 deletion fragments (*Figure 1D–E*). Since PHF13 lacks a predicted DNA binding domain, no DNA sequence specificity could be inferred. Therefore the EMSA's were performed using two random and unrelated DNA fragments, a 248 bp DNA (*Figure 1D*) or a 40 bp DNA fragment (*Figure 1E*). Increasing amounts of GST-PHF13, GST-ΔPHD and ACF1 (positive control) strongly retarded the electrophoretic mobility of the DNA, in contrast to GST alone (*Figure 1D*), indicating that PHF13 can directly interact with DNA independent of its PHD domain. Consistent with these and our previous observations, we mapped the DNA binding region to the middle 100aa (101–200) of PHF13, whereas the N- and C-terminal 100 amino acids of PHF13 did not shift the DNA (*Figure 1E*). Together, these findings support that PHF13 can directly contact chromatin via a direct interaction with DNA.

### PHF13's PHD domain is a specific H3K4me2/me3 reader

The inability of PHF13's PHD domain to interact with recombinant mono-nucleosomes which are devoid of histone post-translational modifications, suggested that it may selectively interact with modified histone tails. Consistently, previous mass spectrometry and bioinformatic predictions indicated that PHF13 may interact with H3K4me3 (*Nikolov et al., 2011*; *Ruthenburg et al., 2007*;

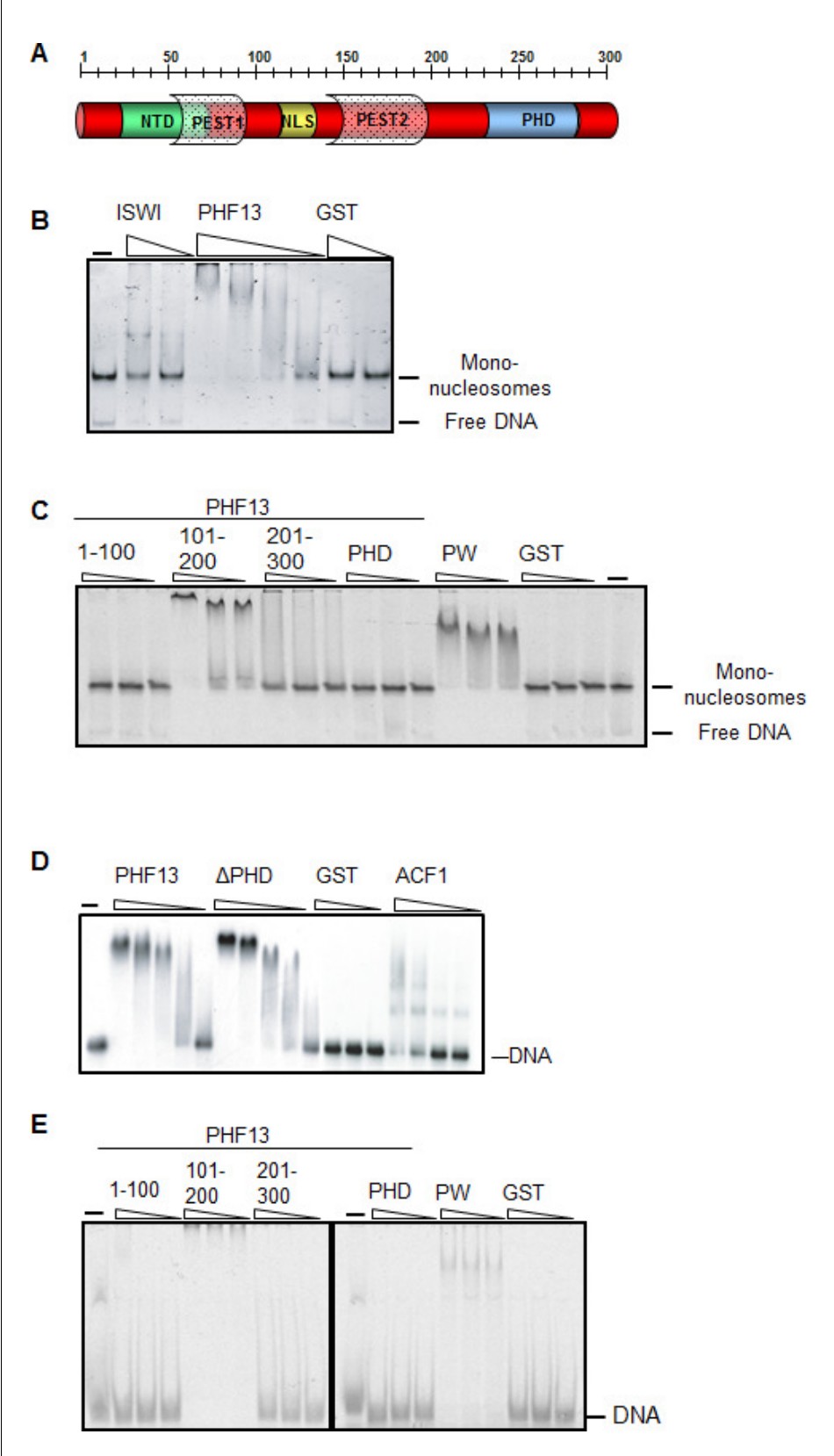

**Figure 1.** PHF13 binds to DNA and recombinant nucleosomes. (**A**) Schematic of the putative domain structure of PHF13. (**B**) Mononucleosome EMSA using recombinant mononucleosomes reconstituted on a 200 bp DNA fragment (20 nM) and increasing concentrations of GST (150 nM, 1200 nM), GST-

*Figure 1 continued on next page*

*Figure 1 continued*

ISWI (28 nM, 226 nM) and GST-PHF13 (70 nM, 140 nM, 280 nM, 560 nM). (**C**) Mononucleosome EMSA using recombinant mononucleosomes reconstituted on a 151 bp fragment (20 nM) and increasing concentrations (80, 160, 320 nM) of GST-1-100, GST-101-200, GST-201-300, GST-PHD and PWWP (positive control). (**D**) EMSA: 248 fM of $P^{32}$ radioactively labeled 248 bp DNA with increasing concentrations of GST (37.5, 150, 300 nM) GST-PHF13 (17, 34, 68, 102 and 135 nM) and GST-ΔPHD (18.5, 37, 74, 111, 148 nM) and ACF1 (10.5, 21, 31.5 and 42 nM). ACF1 served as a positive control. (**E**) EMSA: 10 nM Cy5 labeled 40 bp DNA with increasing concentrations (40, 80, 160 nM) of GST, GST-1-100, GST-101-200, GST-201-300, GST-PHD and GST-PWWP (positive control). The input DNA and mononucleosomes are indicated in B–E.

*Slama and Geman, 2011*). To address this possibility, we evaluated the ability of full-length GST-PHF13 and different deletion fragments to directly interact with either non-modified or differentially methylated histone peptides or with H3K4me3 from nuclease digested chromatin lysate (*Figure 2*). As postulated both GST-PHF13 and the GST-PHF13 fragment containing the PHD domain (201–300) co-precipitated with the H3K4me1, H3K4me2 and H3K4me3 peptides but not with the un-modified H3K4me0 peptide (*Figure 2A*), explaining the inability of the PHD domain to interact with and shift the recombinant mono-nucleosomes. In contrast, GST-only and the GST-PHF13 fragments lacking the PHD domain (1–100 and 101–200) were incapable of co-precipitating with the biotinylated histone peptides. These data strongly support that PHF13's PHD domain interacts with methylated histone H3K4 and suggest that it has a stronger binding preference for H3K4me2 and H3K4me3, in comparison to H3K4me1 (*Figure 2A*).

To further evaluate the specificity of PHF13's PHD domain for H3K4me2/3 in relation to other methyl lysine residues and to calculate approximate binding constants we employed fluorescence polarization analysis using the PHD domain only of PHF13 and differentially modified histone H3 peptides (*Figure 2B*). PHF13's PHD domain bound strongest to H3K4me3 (Kd = 88.5 ± 20.6 μM), with a slightly reduced affinity to H3K4me2 (Kd = 122 ± 29.4 μM) and with a ~4-fold reduced affinity to H3K4me1 (Kd = 332.4 ± 82.6 μM). In contrast to H3K4, no detectable binding was observed for other tri-methylated histone peptides demonstrating a strong preference and specificity of the PHF13 PHD domain for H3K4me2/3 (*Figure 2B*). Finally, to explore whether PHF13's PHD domain can interact with H3K4me3 in a native chromatin context, we analyzed the ability of the different recombinant PHF13 proteins to precipitate H3K4me3 from nuclease digested Hela chromatin lysates (*Figure 2C*). These experiments showed that PHF13 and the deletion fragment containing its PHD domain (201–300) were capable and sufficient to precipitate H3K4me3, an interaction that was lost by deletion of the PHD domain, specific point mutations in the PHD domain (M246A and W255A) that are predicted to disrupt the PHD domain structure, and in fragments not containing the PHD domain (1–100 and 101–200). Together these findings demonstrate that PHF13's PHD domain is a specific H3K4me2/3 molecular reader and together with its DNA binding ability that PHF13 can interact with chromatin in a direct and multivalent manner.

## 3-D structural crystallography of PHF13's PHD domain and H3K4me3

To gain structural insight into this specific recognition of histone H3K4me3 by the PHD domain of PHF13, we determined the crystal structures of both the apo-PHF13 PHD domain (aa 226–280) and the PHF13 PHD-H3K4me3 peptide complex (*Figure 3A-C*). The diffraction data and refinement statistics for these structures can be found in *Figure 3—source data 1*. Similar to other PHD domains (*Adams-Cioaba and Min, 2009*; *Musselman and Kutateladze, 2009*), the PHD domain of PHF13 folds into a canonical Cys4-His-Cys3 (or C4HC3) motif, that coordinates two zinc ions with a well-conserved globular domain (*Figure 3A*). In the complex structure of PHF13 PHD-H3K4me3, we can see that the histone H3K4me3 peptide is bound against the central double-strand anti-parallel β-sheet of the PHF13 PHD domain and completes a three-stranded β-sheet (*Figure 3A*). The first four residues of the H3K4me3 peptide are embedded in an exposed binding groove on the surface of the PHF13 PHD domain (*Figure 3B–C*). Similar to other methyl lysine binding proteins (*Adams-Cioaba and Min, 2009*), the trimethyl lysine 4 (K4me3) is bound in an aromatic cage formed by residues F241, M246 and W255 of PHF13 (*Figure 3B–C*). In addition, the K4me3 residue also forms two main chain hydrogen bonds with M246 (*Figure 3C*). The first residue alanine (Ala1) of the H3K4me3 peptide is anchored in a small and secluded pocket created by I247, V268, P269, E270 and F272, and the backbone amine group of Ala1 forms a hydrogen bond with the backbone carbonyl oxygen

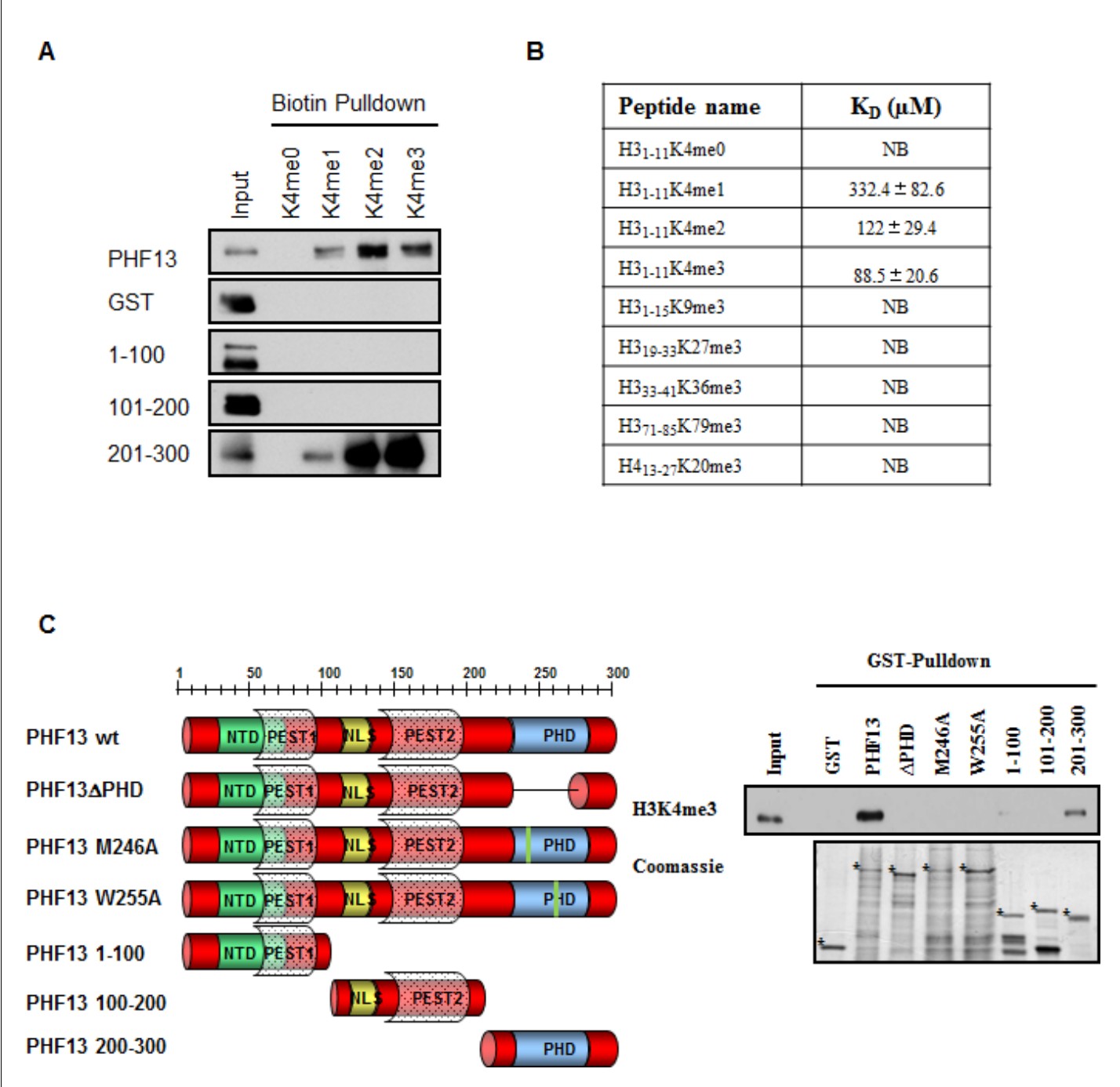

**Figure 2.** PHF13's PHD domain specifically interacts with H3K4me2/3 (**A**) Biotinylated histone peptide pull down.Equivalent amounts (1 μg) of GST, GST-PHF13 and GST-PHF13 deletion fragments 1–100, 100–200 and 200–300 aa were incubated with 1 μg of differentially modified biotinylated histone peptides and streptavidin beads. Co-precipitation of GST-proteins was analyzed using a GST specific antibody. (**B**) Fluorescence Polarization Assay. The dissociation constant of the PHF13-PHD only protein with differentially methylated H3 and H4 peptides. (**C**) GST pull down of H3K4me3 from nuclease digested chromatin lysates using GST-alone, GST-PHF13 and the indicated GST-PHF13 deletion and point mutant proteins. Precipitation of H3K4me3 was analyzed with a specific antibody. Amount of GST proteins were controlled by Coomassie staining.

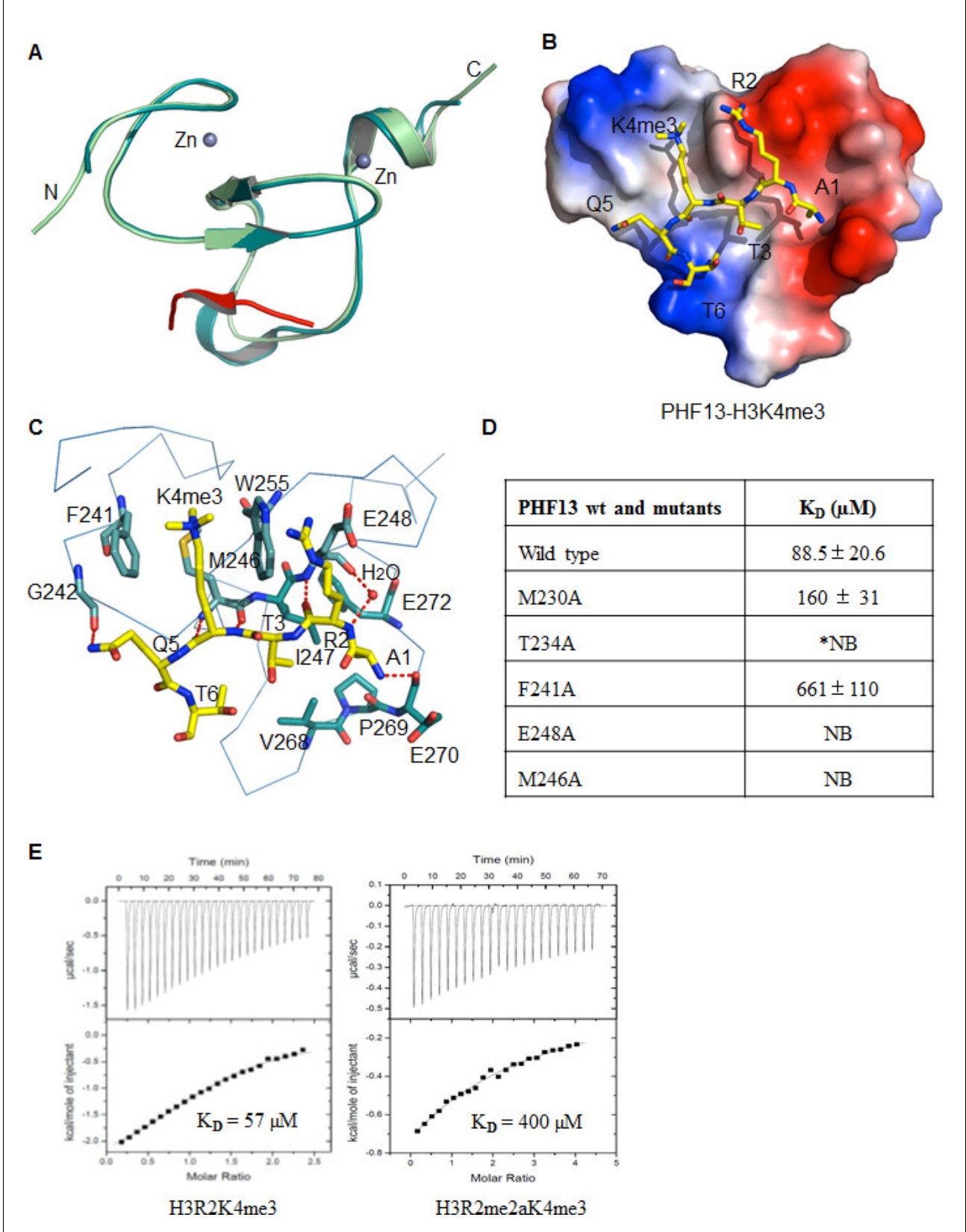

**Figure 3.** Crystal structure of PHF13's PHD domain. (**A–B**) The crystal structure of apo-PHF13 PHD domain (**A**) and PHF13 PHD finger in complex with H3K4me3 (**B**). (**C**) Electrostatic surface potential of PHF13's PHD finger in complex with H3K4me3. Dashed lines represent intermolecular hydrogen

*Figure 3 continued on next page*

*Figure 3 continued*

bonds. (D) Fluorescence Polarization Assay - The dissociation constant of PHF13 and PHF13 PHD point mutants with H3K4me3. (E) Isothermal titration Calorimetry (ITC) - Binding affinity of PHF13's PHD domain for H3R2K4me3 (left panel) or H3R2me2aK4me3 (right panel).

The following source data and figure supplement are available for figure 3:

**Source data 1.** Data collection and refinement statistics – Detailed specifications of the data obtained from the crystallization of Apo-PHF13-PHD and the PHF13-PHD-H3K4me3 complex.

**Source data 2.** PHF13 PHD domain binding to a differentially methylated histone peptide tail chip.

**Figure supplement 1.** Comparison of the H3R2 binding pocket of different PHD domains.

of P269 and E270 in PHF13 (*Figure 3C*). The limited length between the aromatic cage and the Ala1 binding pocket and the restricted nature of the Ala1 binding pocket determines that the PHF13 PHD domain selectively recognizes only methylated histone H3K4. Consistently, PHF13 PHD domain was not co-precipitated by biotinylated H3K4me0 (*Figure 2A*), nor did it interact with recombinant mononucleosomes (*Figure 1B*) or H3K4me0 fluorescent peptides (*Figure 2B*). The Arg2 residue in the H3K4me3 peptide is bound by two backbone hydrogen bonds with E248 (*Figure 3B–C*). Mutation of any of the residues affiliated with the H3K4me3 or H3R2 binding pockets (T234A, F241A, M246A, W255A and E248A) abrogated binding of PHF13's PHD domain to H3K4me3 (*Figure 2C* and *3D*).

The surface charge representation shows that Arg2 sits in a shallow negatively charged pocket, reminiscent of the Arg2 binding in the ING2 and BPTF2 PHD domain-H3K4me3 complex structures, but in contrast to the RAG2-H3K4me3 complex structure, in which Arg2 is bound in a relatively hydrophobic pocket (*Figure 3C* and *Figure 3—figure supplement 1*). This indicates that similar to ING2 and BPTF, that Arg2 methylation should diminish PHF13 binding to H3K4me3, whereas it has been shown to enhance RAG2 binding (*Iberg et al., 2008*; *Yuan et al., 2012*). Therefore to address this prediction and test the influence of Arg2 methylation on PHF13's PHD domain H3K4me3 binding ability we performed isothermal titration calorimetry (ITC; *Figure 3E*). Our binding results show that the binding of PHF13's PHD domain to H3K4me3 was diminished by the simultaneous asymmetric dimethylation of Arg2 (H3R2me2aK4me3). Consistently, the binding of PHF13's PHD domain to H3K4me3 on a differentially modified histone peptide chip was also significantly reduced in the presence of dimethyl Arg2 (*Figure 3—source data 2*). Together, these results confirm that the PHD domain of PHF13 is an H3K4me3 reader and demonstrate the structural relationship between them. Furthermore, they identify key residues within PHF13's PHD domain involved in mediating this molecular interaction as well as inhibiting modifications of neighboring histone residues that opposes their interaction.

## PHF13 interacts with PRC2 and RNA polymerase II

To gain additional information about PHF13 chromatin interactions, we immunoprecipitated PHF13 from the nuclease digested chromatin fraction of mESCs and analyzed the co-immunoprecipitating proteins in comparison to an IgG control by mass spectrometry (*Figure 4A* and *Figure 4—source data 1*). Independent PHF13 IPs were digested by trypsin or LysC (*Figure 4—figure supplement 1*) prior to MS analysis and the co-precipitating proteins were combined (*Figure 4A* and *Figure 4—source data 1*). Similar complexes were retrieved regardless of whether the results were pooled (*Figure 4A* and *Figure 4—source data 1*) or examined individually (*Figure 4—figure supplement 1*). For all interactions where only a single unique peptide was detected the mass spectrometry profile is shown, to validate their identity (*Figure 4—figure supplement 2–4*). The mass spectrometry findings confirmed previous observations that PHF13 interacts with DNA damage response (DDR) proteins, ATM and TRIM 28 (*Mund et al., 2012*) and revealed that PHF13 interacts with an RNA polymerase II/ splicing complex as well as with several components of PRC2, namely, SUZ12, RBBP4 and RBBP7 (*Figure 4A*). Analysis of the associated functional terms affiliated with PHF13 interacting proteins, indicated a role in RNA metabolic processes, gene expression, RNA and nucleic acid binding (*Figure 4—source data 2*). To reproduce these observations we performed reciprocal co-

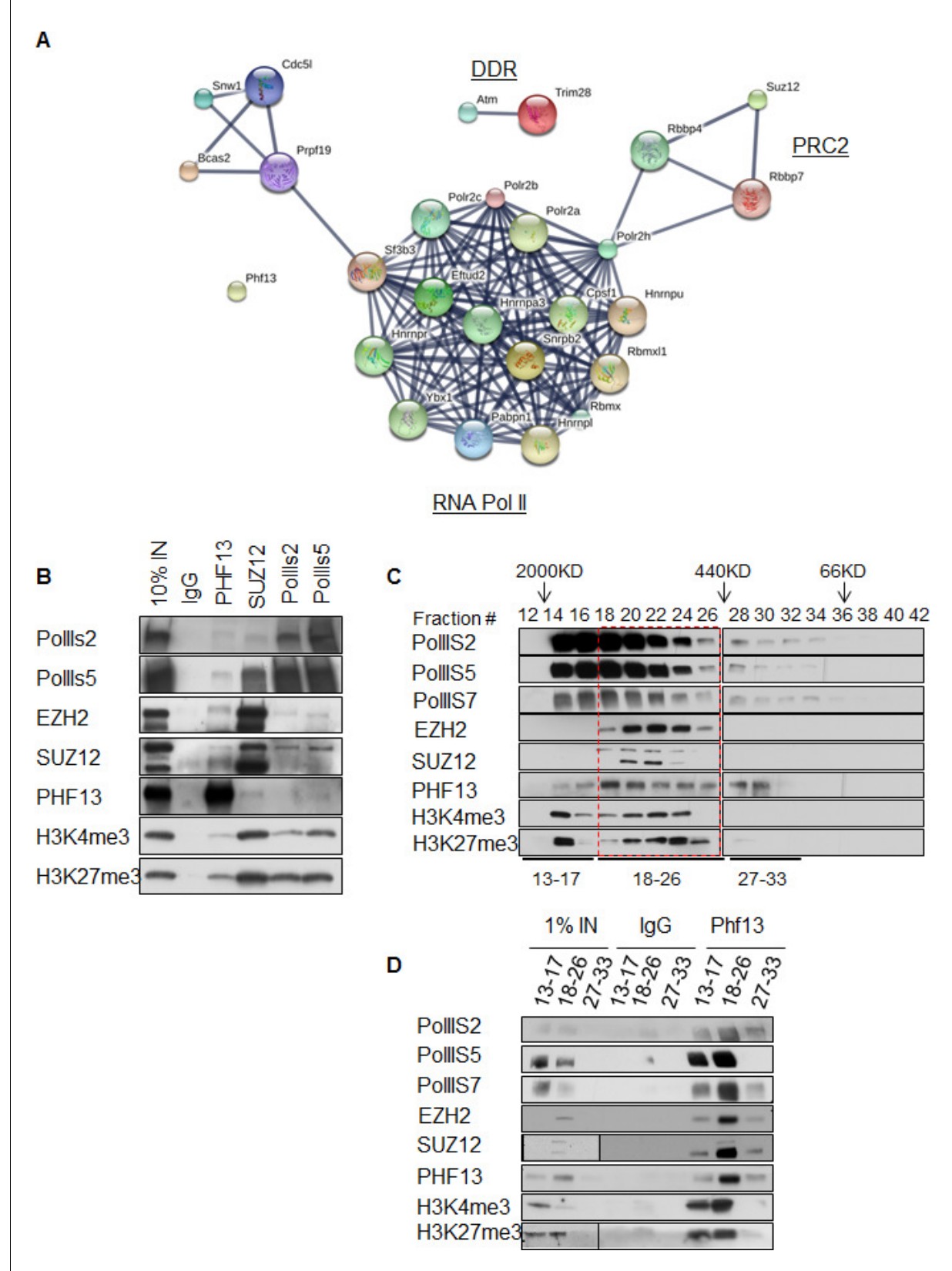

**Figure 4.** PHF13 interacts with RNA polymerase II and PRC2 complexes. (**A**) A String functional protein association network (http://string-db.org/) of PHF13 chromatin interactions as identified by mass spectrometry. High confidence interactions were selected (0.9), and revealed an interaction network

*Figure 4 continued on next page*

*Figure 4 continued*

with DDR proteins, PRC2 and RNA polymerase II complex. (**B**) Immunoblot of co-immunoprecipitating interactions from control IgG, PHF13, SUZ12, PolII S2P and PolII S5P immunoprecipitations from nuclease digested chromatin fraction of E14 mESC cells. (**C**) Immunoblot of even fraction numbers collected from a superose 6 size exclusion chromatography run using nuclease digested chromatin fraction of E14 mESC cells. The co-elution of RNA PolII with PRC2, H3K4me3, H3K27me3 and PHF13 in fractions 18–26 is denoted by red hatched box. (**D**) Immunoblot of interacting proteins from an IgG and PHF13 immunoprecipitation of the pooled chromatography fractions 13–17, 18–26 and 27–33. Inputs for SUZ12 and H3K27me3 are separately boxed due to the fact that they represent a separate exposure in relation to the IPs.

The following source data and figure supplements are available for figure 4:

**Source data 1.** Mass spectrometry table of PHF13 interacting proteins.
**Source data 2.** Associated functional terms of mass spectrometry interacting proteins.
**Figure supplement 1.** String protein functional association networks for PHF13 interactions when digested by Trypsin (**A**) or by LysC (**B**).
**Figure supplement 2.** Mass spectrometry profiles of PHF13 interacting proteins where only one unique peptide was identified.
**Figure supplement 3.** Mass spectrometry profiles of PHF13 interacting proteins where only one unique peptide was identified.
**Figure supplement 4.** Mass spectrometry profiles of PHF13 interacting proteins where only one unique peptide was identified.

immunoprecipitation in mESCs which confirmed that PHF13, SUZ12 and RNA PolII S5P interact with each other and that they all co-precipitate with H3K4me3 and H3K27me3 (*Figure 4B*). To provide additional evidence that these proteins exist in a common chromatin molecular complex we performed size exclusion chromatography from nuclease digested chromatin fraction of mESCs to look for fractions where PHF13 coexists with PRC2, RNA PolII, H3K4me3 and H3K27me3 (*Figure 4C*). PHF13 was present in three different compositions, a very high molecular weight fraction (fractions 14–16) where it co-eluted with RNA PolII (S2P, S5P and S7P) H3K4me3 and H3K27me3, a high-mid molecular weight fraction (fractions 18–26) where it co-eluted with RNA PolII (S2P, S5P and S7P), PRC2 (EZH2 and SUZ12), H3K4me3 and H3K27me3 and in mid-low molecular weight fractions (28–32) where it co-eluted only with RNA PolII (*Figure 4C*). Together, these findings raise the possibility that PHF13 co-exists with different RNA PolII complexes, i.e. some containing PRC2 and histones and some that do not. To address this possibility we pooled the very high (13–17), high-mid (18–26) and mid-low (27–33) fractions and examined whether PHF13 could co-precipitate RNA PolII, PRC2, H3K4me3 and H3K27me3 from these different fractions (*Figure 4D*). PHF13 interacted with RNA PolII S2P and S7P in all three pooled fractions (13–17, 18–26 and 27–33) where as it interacted with RNA PolII S5P, H3K4me3 and H3K27me3 only in the larger molecular weight fractions (13–17 and 18–26). Furthermore, PRC2 predominantly co-precipitated with RNA PolII, PHF13, H3K4me3 and H3K27me3 in fractions 18–26. Together, these findings reveal that PHF13 interacts with RNA PolII in different complex constellations and support the idea that PHF13 may play a role in gene regulation.

## PHF13 overlaps with H3K4me2/3, PRC2, RNA PolII, DHS sites and CpG islands

Our results strongly suggest that PHF13 binds nucleosomes marked by H3K4me2/3 via its C-terminal PHD domain – an interaction that can be further stabilized by PHF13's ability to bind DNA and chromatin affiliated PRC2 and RNA PolII. Thus, we expect that PHF13 should co-localize with H3K4me2/3, PRC2 (SUZ12 and EZH2), RNA PolII and certain DNA sequence motifs in vivo and genome wide. To test this idea we identified PHF13 bound regions by chromatin immunoprecipitation followed by sequencing (ChIPseq) and quantified their overlap with regions marked or bound by H3K4me1, H3K4me2, H3K4me3, H3K9me3, H3K27me3, SUZ12, EZH2 and the different hypophosphorylated and phosphorylated forms of RNA PolII (*Figure 5*) obtained from publicly available ChIPseq datasets for mESCs (Materials and methods). We identified 17,937 PHF13 bound regions and confirmed a few of them by ChIP qPCR (*Figure 5—figure supplement 1A*). PHF13 bound regions strongly

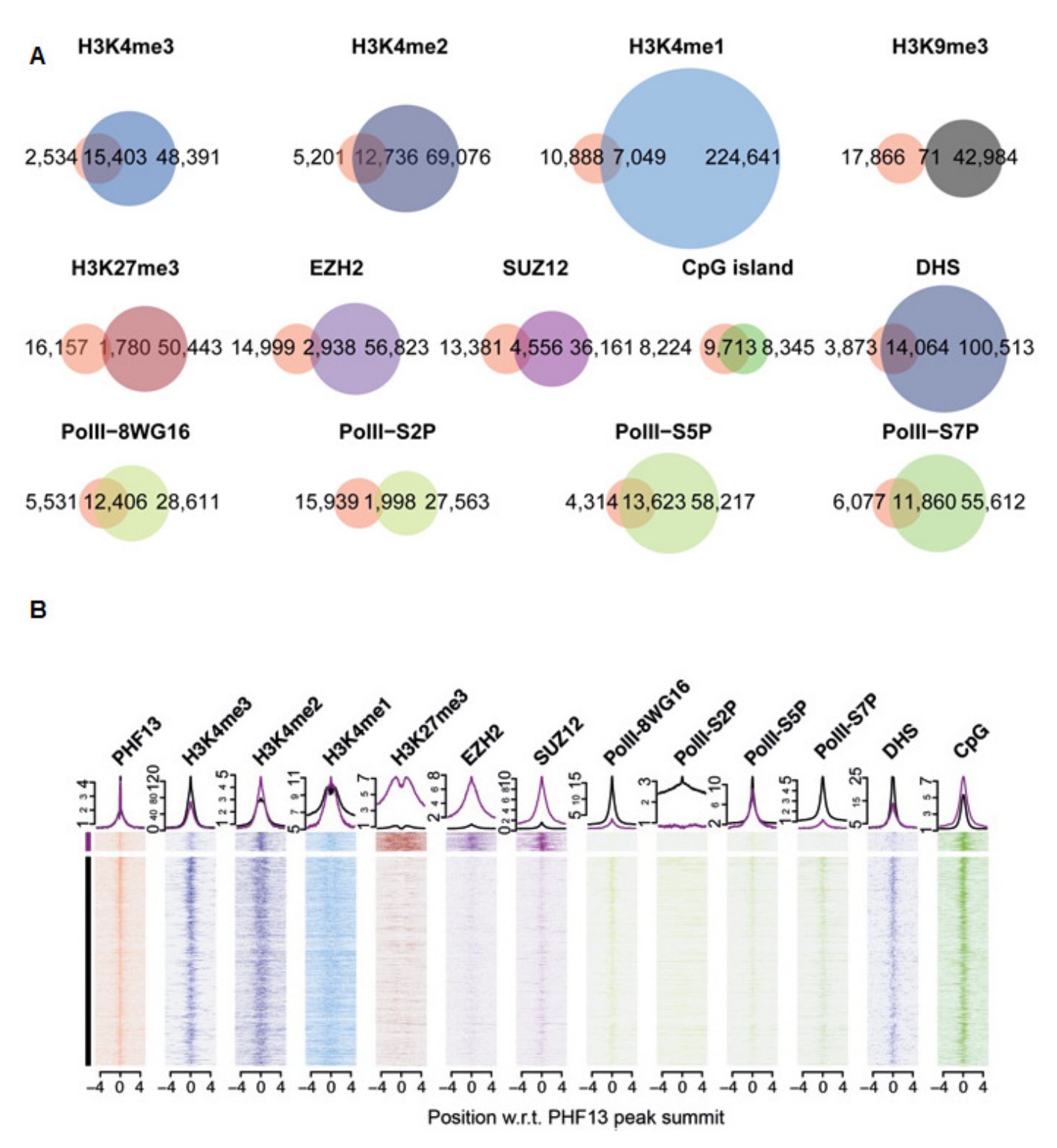

**Figure 5.** ChIPseq shows a genome wide overlap with methylated H3K4, DHS, CpG islands, PRC2 and RNAPolII. (**A**) Venn diagrams for the overlap of called-peaks. (**B**) Heatmaps for the ChIPseq signal centered around PHF13 peaks. Shown are two groups of peaks that are the result of a k-means clustering of all ChIPseq signals except the one from PHF13, the DHS signal, and the CpG content. Above the heatmap the average signal for the two groups is plotted.

*Figure 5 continued on next page*

*Figure 5 continued*

The following figure supplements are available for figure 5:

**Figure supplement 1.** ChIP qPCR analysis of PHF13 ChIPseq targets.

**Figure supplement 2.** RFAT motif finder of PHF13 ChIP sequencing peak.

overlap with H3K4me3 (87%) and H3K4me2 (71%), whereas this overlap was decreased with H3K4me1 (39%; *Figure 5A*). Combined with the biochemical and structural findings, these observations support that PHF13 is a *bona fide* H3K4me2/3 reader.

Similarly, we examined the overlap of PHF13 bound regions with H3K9me3 and H3K27me3 repressive modifications (*Figure 5A*). Not surprisingly, we observed essentially no overlap with H3K9me3, a mark that is mutually exclusive to H3K4me3 in mESCs except at imprinted genomic regions (*Dindot et al., 2009*). In contrast however, we did find a modest overlap with H3K27me3 repressive marks (10%). Similar to H3K27me3 we observed a modest but relevant overlap of PHF13 peaks with EZH2 (16%) and SUZ12 (25%) whereas 5% of EZH2 and 11% of SUZ12 peaks overlapped with PHF13. These findings demonstrate that PHF13 also co-localizes at a subset of Polycomb demarcated loci and is consistent with PHF13 interacting with PRC2.

Next we examined the overlap of PHF13 ChIPseq peaks with the hypophosphorylated, serine 2, serine 5 and serine 7 phosphorylated forms of RNA PolII (*Figure 5A*). We observed a substantial overlap between PHF13 targets and hypophosphorylated (69%), serine 5 phosphorylated (76%) and serine 7 phosphorylated (66%) forms of RNA PolII, whereas the serine 2 phosphorylated form only showed a modest overlap (11%). These observations argue that PHF13 interacts with promoter affiliated RNA polymerase II and not the elongating form, consistent with the fact that PHF13 is enriched at H3K4me3 (promoter mark) and not with H3K36me3 (found in the gene body).

Finally, given the in vitro DNA binding activity of PHF13 (*Figure 1*), we also asked the question whether PHF13 bound regions coincide with DNAse I hypersensitive (DHS) sites (*Figure 5A*). We observed a significant overlap between PHF13 targets and DHS sites (78%) which is consistent with an in vivo DNA binding activity to nucleosome free and/or complexed DNA. To further discern whether PHF13 recognizes specific DNA motifs, in silico analysis was performed using RSAT Peak motifs (*Thomas-Chollier et al., 2012*). This analysis found that PHF13 bound regions were enriched at CpG rich motifs and depleted at AT rich sequences (*Figure 5—figure supplement 2*). This observation prompted us to additionally quantify the overlap of PHF13 targets with annotated CpG islands (*Figure 5A*), which similarly revealed a significant overlap (~54%). Together, these findings implicate CpG rich DNA in PHF13 chromatin recruitment and/or stabilization at H3K4me2/3. The relationship of PHF13 at active and repressed promoters was further visualized in the UCSC genome browser (*Figure 5—figure supplement 1B*).

To better understand these PHF13 chromatin contexts we employed the k-means clustering approach to separate PHF13 bound regions by the patterns of histone modifications and chromatin binding proteins (*Figure 5B*; see Materials and methods for details). The results of this analysis (*Figure 5B*) show two distinct regimes. The first regime is a group of PHF13 bound regions containing H3K4me2/3, H3K27me3, PRC2 (SUZ12 and EZH2), RNA PolII S5P, CpG enrichment and is depleted of RNA PolII S2P and DNase I hypersensitivity. The absence of RNA PolII S2P and DNase I hypersensitivity and the presence of PRC2 and H3K27me3 is consistent with these regions being not accessible and repressed. Interestingly, this regime is reminiscent of bivalent promoters which are demarcated by the presence of both H3K4me3 and H3K27me3, Polycomb, high CpG content and specifically the serine 5 phosphorylated form of RNA polymerase II (*Brookes et al., 2012*; *Lynch et al., 2012*; *Orlando et al., 2012*; *Thurman et al., 2012*; *Wachter et al., 2014*). Furthermore, it is worthy to note that bivalent promoters represent only a small fraction of H3K4me3 and H3K27me3 positive promoters, and might reflect the approximate 10% representation observed here. The second larger group contains H3K4me2/3, RNA PolII S2P, RNA PolII S5P, DHS sites, CpG enrichment and a notable absence of H3K27me3 and PRC2, indicating that these regions are accessible and expressed (*Antequera and Bird, 1999*; *Orlando et al., 2012*; *Tazi and Bird, 1990*; *Thurman et al., 2012*). In all cases, with the exceptions of H3K27me3 and RNA PolII S2P, the levels

peak at the position where PHF13 levels are highest, indicating a co-localization of the signals. These findings indicate that CpG rich sequences, RNA PolII S5P and H3K4me2/3 coexist at PHF13 targets and therefore likely cooperate in PHF13 recruitment and/or function at active and repressed chromatin states.

## PHF13 is a transcriptional co-regulator

To gain insight about the functional role of genes targeted by PHF13, we performed an overrepresentation analysis of associated functional terms. We defined PHF13 target genes (10,826) by intersecting a window of +/- 1500 base pairs around their transcription start sites (TSS) with PHF13 bound regions (*Figure 6—source data 1*). The overrepresentation analysis revealed that PHF13 targets are functionally enriched in transcription, cell cycle, DNA repair, chromatin organization and developmental processes (*Figure 6A* and *Figure 6—source data 2*) consistent with previous reports on PHF13 functions (*Bördlein et al., 2011*; *Kinkley et al., 2009*; *Mund et al., 2012*).

Since PHF13 interacts with RNA PolII in different constellations, with and without PRC2 (*Figure 4*) and since the ChIP sequencing results indicated that PHF13 localized to the promoters of both active and repressed genomic regions (*Figure 5*), we suspected that PHF13 may have a transcriptional co-regulatory function and that its depletion in mESCs may positively or negatively influence gene expression. To address this possibility we performed RNAseq on mESCs that were transduced with a PHF13 specific shRNA or a scrambled shRNA control (*Figure 6B and C*). The efficiency of the knockdown was controlled 12 days later by qPCR and western blot and showed an approximate 80% reduction (*Figure 6—figure supplement 1*). Differential gene expression analysis showed a total of 1386 genes (*Figure 6—source data 3*) that were up or down regulated at an adjusted p-value less than 0.05. Of these 807 genes went down and 579 genes went up after PHF13 depletion, supporting the idea that PHF13 can promote both gene activation and repression. Several up and down regulated genes identified by RNA sequencing were confirmed by qPCR in control and PHF13 shRNA depleted E14 mESCs and in a Doxyclin inducible PHF13 shRNA mESC cell line that expresses a completely independent shRNA (*Figure 6—figure supplement 1*). Of the 1386 genes that were affected by PHF13 depletion, 845 genes were also identified as PHF13 targets by the ChIPseq (*Figure 6B*) supporting a direct correlation between the occupancy of PHF13 at these targets and their expression levels. Of these 845 genes, 487 were upregulated and 358 were downregulated. Gene set enrichment analysis of the genes affected by PHF13 depletion further revealed a similar overrepresentation of associated functional terms to ChIPseq and mass spectrometry targets and identified transcription, cell cycle, chromosome organization, differentiation, DNA binding, RNA binding and chromatin binding (*Figure 6C* and *Figure 6—source data 4* and *5*). Together these findings support that PHF13 interacts with transcriptional regulatory proteins, that it binds to genes influencing transcription, cell cycle, chromosome organization and differentiation and that its depletion alters genes with similarly associated functional terms, all of which is consistent with a transcriptional co-regulatory role of PHF13.

## PHF13 depletion leads to upregulation of Polycomb repressed genes and the down regulation of active genes

To further explore the relationship between the differentially expressed genes and the ChIPseq levels of PHF13, H3K4me1, H3K4me2, H3K4me3, H3K27me3, EZH2, SUZ12, and the differently phosphorylated forms of RNA PolII we compared their normalized levels to the corresponding levels in the unchanged genes (*Figure 7A*). We found that the genes that were both up- and down- regulated had significantly higher PHF13 enrichment than the genes that were unchanged (pvalue < 0.0001; Wilcoxon rank sum and signed rank test) reflecting a causal relationship of PHF13 binding and function. Furthermore, both the genes that were up- and down- regulated had significantly higher levels of H3K4me1, H3K4me2, H3K4me3 and RNA PolII and significantly lower levels of H3K27me3 in comparison to the unchanged genes. This reflects the fact that PHF13 target genes are transcriptionally more active than the non-changed genes which are predominantly repressed and is consistent with PHF13 specificity for H3K4me2/3. The genes that decrease in expression after PHF13 depletion had significantly more H3K4me3 and RNA PolII S2P than the genes that increased after PHF13 depletion, indicating that they are highly expressed. In contrast, the genes that increase after PHF13 depletion have significantly higher levels of H3K4me1, H3K4me2, H3K27me3, EZH2 and

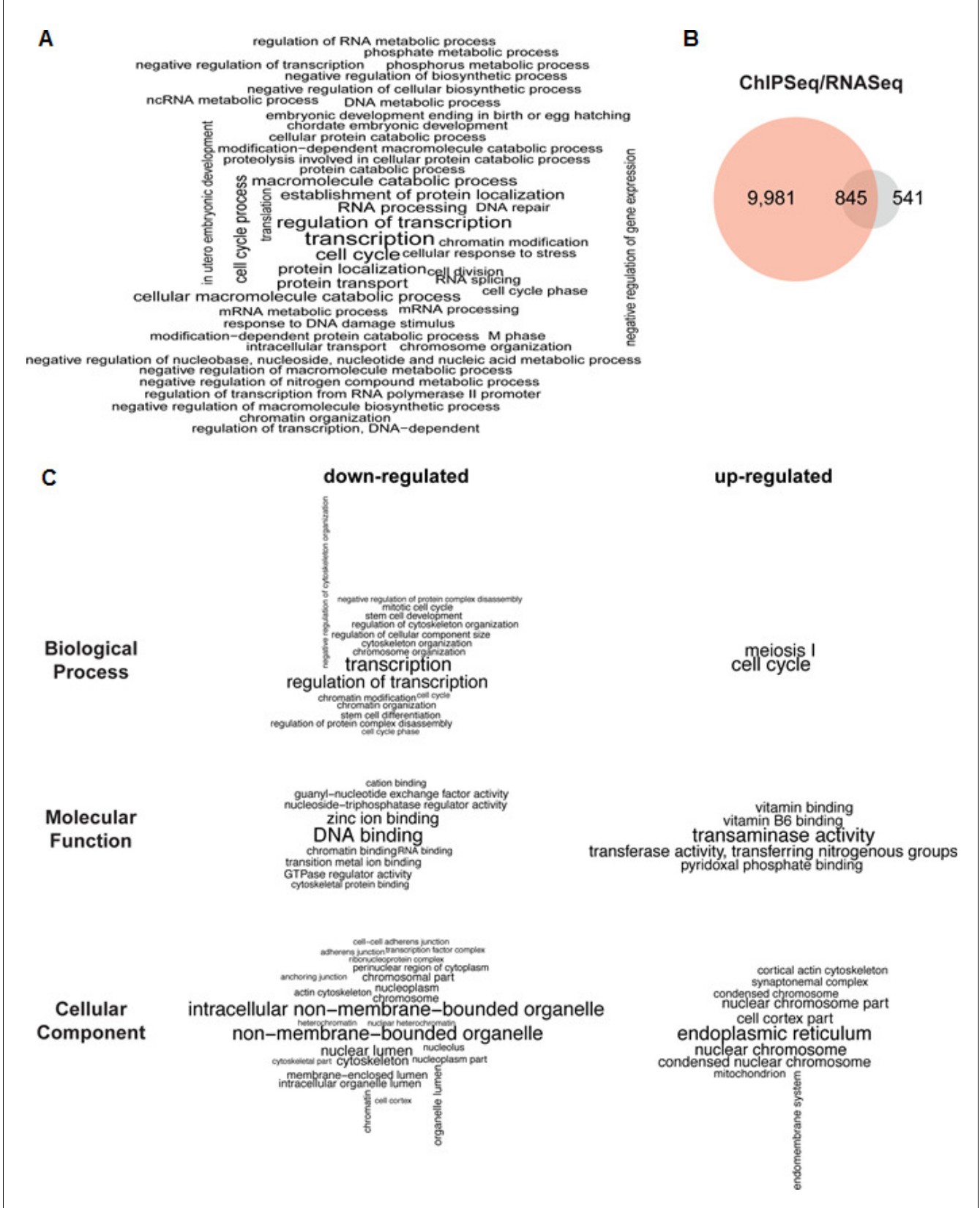

**Figure 6.** PHF13 ChIP and RNA sequencing overlap and associated functional terms. (A) Gene set enrichment analysis of genes overlapping with PHF13 ChIPseq peaks located within +/- 1500 bp of their TSSs. (B) Venn diagrams showing the overlap of PHF13 ChIPseq target genes and the genes

*Figure 6 continued on next page*

*Figure 6 continued*

that were significantly up or down regulated (with an adjusted p-value less than 0.05) after PHF13 knockdown. (**C**) Gene ontology analysis of the significantly differentially expressed genes that were up or down regulated (with an adjusted p-value of less than 0.05) after PHF13 knockdown.

The following source data and figure supplement are available for figure 6:

**Source data 1.** PHF13 ChIPseq targets- PHF13 ChIPseq peaks in mouse ES cells located +/- 1500 bp of the TSSs were used to identify 10,826 PHF13 target genes.
**Source data 2.** David GO analysis of PHF13 ChIPseq targets- PHF13 ChIP sequencing target genes were analysed by david and resulted in the following biological processes, molecular functions and cellular components being over represented.
**Source data 3.** PHF13 shRNA RNAseq targets- PHF13 shRNA depletion for 12 days led to 1,386 genes being significantly up or down with an adjusted p-value less than 0.05 in mouse ES cells.
**Source data 4.** David GO analysis of PHF13 regulated genes- Down regulated genes in mESCs after PHF13 knockdown were analyzed by David functional annotation bioinformatic microarray analysis and returned the following biological processes, molecular functions and cellular components as being over represented.
**Source data 5.** David GO analysis of PHF13 regulated genes- Up regulated genes in mESCs after PHF13 knockdown were analyzed by David functional annotation bioinformatic microarray analysis and returned the following biological processes, molecular functions and cellular components as being over represented.
**Figure supplement 1.** qPCR validation of PHF13 regulated genes.

SUZ12 and significantly less RNA PolII S2P, than the genes that decrease after PHF13 depletion indicating that they are normally silenced. Interestingly, we also observed significantly higher levels of H3K4me3 on these genes in comparison to the non-changed genes, albeit with significantly lower H3K4me3 levels than on the genes that decreased upon PHF13 depletion. We interpret the co-occurrence of H3K4me2/3, H3K27me3 and PRC2 (SUZ12 and EZH2), with high levels of PolII S5P and low levels of PolIIS2P to represent bivalent domains. Consistently, these chromatin features resemble the first PHF13 cluster identified k-means clustering (*Figure 5B*).

The observations that PHF13 interacts with PRC2 and RNA polymerase II, that they co-localize together at both active and inactive gene promoters and that PHF13 depletion alters the gene expression of a subset of these has lead us to propose that PHF13 acts as a transcriptional co-regulator. In an effort to better understand how PHF13 is influencing transcription, we employed a Doxy-cyclin inducible PHF13 shRNA mESC cell line. Using this tool we immunoprecipitated PHF13, SUZ12, RNA PolII S2P and RNA PolII S5P from wild type (- Dox) and PHF13 depleted (+ Dox) nuclease digested chromatin lysates. We then checked whether PHF13 depletion interfered with the co-precipitation of PHF13, SUZ12, RNA PolII S2P, RNA PolII S5P, H3K4me3 and H3K27me3, from each of these immunoprecipitations (*Figure 7B*). As expected PHF13 knockdown (+ Dox) resulted in a reduction in the total levels of PHF13 and in its reduced precipitation and co-precipitation by SUZ12, RNA PolII S2P and RNA PolII S5P. Interestingly, in addition we observed that SUZ12, RNA PolII S2P and RNA PolII S5P also showed reduced co-precipitation of H3K27me3 and that SUZ12 and RNA PolII S5P, but not RNA PolII S2P showed a reduced co-precipitation with H3K4me3 (*Figure 7B*), a finding that is consistent with the fact that RNA Poll S2P is not normally localized at H3K4me3 demarcated promoters. Furthermore, RNA PolII S5P additionally showed a reduced co-precipitation of PRC2 (SUZ12 and EZH2), whereas this interaction was unaffected in RNA PolII S2P (*Figure 7B*). These findings suggest that PHF13 can act as a scaffold stabilizing RNA Poll S5P and PRC2 at bivalent H3K4me3 and H3K27me3 demarcated promoters and similarly stabilizes RNA Poll S5P likely with other transcriptional activating complexes at active H3K4me3 demarcated promoters. These observations indicate an influential role of PHF13 in targeting or stabilizing transcriptional activating and repressing complexes at H3K4me2/3 containing promoters and thereby impacting transcriptional activity.

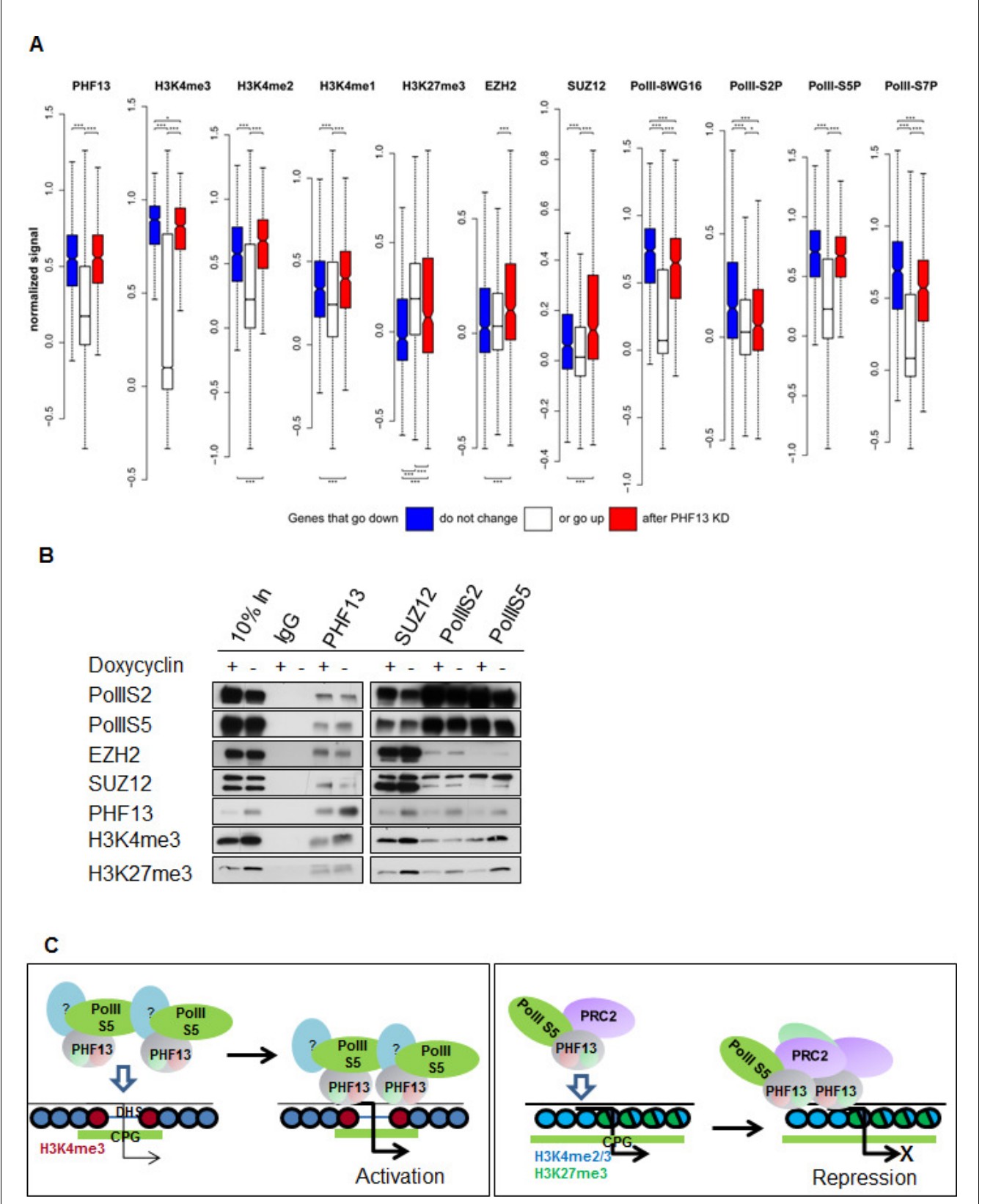

**Figure 7.** PHF13 modulates recruitment and/or stabilizes RNA PolII S5P and RNA PolII S5P / PRC2 complexes at H3K4me1/2/3 demarcated chromatin. (A) PHF13 targets that go up or down after depletion are under wild-type conditions enriched or depleted of PRC2, respectively and are enriched in

*Figure 7 continued on next page*

*Figure 7 continued*

H3K4me1/2/3 and RNA PolII. Differential transcript expression upon PHF13 knockdown in murine ES cells. For each ChIPseq track the normalized signal around the transcription start site is shown for transcripts, which are down-regulated (blue), not changed (white), up-regulated (red) at an adjust p-value threshold of 0.05. The bars above each boxplot denote statistical significant higher signal in the down-/up-regulated genes compared to the unchanged transcripts and in the down regulated genes compared to the up regulated genes. The bars below each boxplot denote statistical significant lower signals in the down-/up-regulated genes compared to the unchanged transcripts and in the down regulated genes compared to the up regulated genes. The stars above and below the bars denote the significance threshold, which is for one star 0.01, two 0.001 and three 0.0001. P-values were calculated with the Wilcoxon rank sum test. (B) Immunoblots of co-precipitating proteins from IgG, PHF13, SUZ12, PolIIS2 and PolIIS5 immunoprecipitations performed in nuclease digested chromatin lysates from wild type (- Doxycyclin) or PHF13 depleted (+ Doxycyclin) mESCs. (C) **Left Panel:** PHF13 interacts with RNA PolII (S5P and S7P) and most probably as of yet unknown transcriptional activating chromatin modulating complexes. Furthermore, PHF13 simultaneously binds to H3K4me3 and CpG rich DNA at the promoters of active genes. This in turn, acts to either recruit and/or stabilizes these complexes at active promoters, promoting transcription. **Right Panel:** PHF13 interacts with RNA PolII S5P and PRC2. PHF13 simultaneously binds to bivalent CpG rich, H3K4me2/3 and H3K27me3 enriched promoters of silenced genes. This in turn acts to recruit and/or stabilizes PRC2 and RNA PolII S5P at bivalent promoters, thereby promoting gene repression.

## Discussion

In this manuscript we demonstrate that PHF13 is a novel H3K4me2/3 reader and effector. We show that its affiliation with H3K4me2/3 and CpG rich DNA sequences facilitates its specific recruitment to repressed or active chromatin regions, interactions that are conceivably strengthened by additional interactions with PRC2 and RNA PolII. PHF13 depletion disrupts the interactions between PRC2, RNA PolII S5P, H3K4me3 and H3K27me3, indicating that they are in common complexes and that PHF13 is integral to their localization at active and bivalent promoters. Furthermore, PHF13 depletion alters the gene expression of a subset of genes affiliated with these proteins consistent with a co-transcriptional role.

The PHD domain of PHF13 displays a specific albeit weak binding affinity to H3K4me2/3, a phenomenon that is common to many PHD domains (*Musselman and Kutateladze, 2009*; *Musselman and Kutateladze, 2011*) and indicates that additional chromatin interactions are required for this interaction to be stable under physiological conditions. In line with this, we show that PHF13 can directly affiliate with nucleosome free or complexed DNA via a centrally located domain and with chromatin affiliated PRC2 and RNA PolII, arguing that PHF13 contacts chromatin in a multivalent fashion. The ability to simultaneously bind DNA, H3K4me2/3 and other chromatin affiliated proteins increases PHF13's binding avidity for chromatin, thereby stabilizing weak H3K4me2/3 interactions. Such a feature is presumably beneficial for PHF13's ability to recognize DNA damage and perform its DNA repair functions (*Mund et al., 2012*) as well as its ability to impact transcription.

Analysis of PHF13 ChIPseq targets identified that PHF13 localizes to at least 2 distinct chromatin landscapes (*Figure 5C*). A Polycomb enriched chromatin environment and an H3K4me3 active chromatin state. Interestingly, PHF13 ChIPseq peaks overlapped predominantly with H3K4me2 (and to a lesser extent with H3K4me3) in Polycomb enriched chromatin and predominantly with H3K4me3 (and to a lesser extent with H3K4me2) at active promoters. This sub-clustering of PHF13 peaks may indicate that PHF13's functional impact on chromatin depends on whether it interacts with H3K4me2 or H3K4me3. Consistently, genes that showed increased expression after PHF13 depletion also showed the highest levels of H3K4me2, H3K27me3 and Polycomb, in comparison to those that decreased after PHF13 depletion which showed the highest levels of H3K4me3 and RNA PolII. In addition, in silico DNA motif analysis performed on PHF13 ChIP sequencing peaks revealed that it is affiliated with CpG rich DNA and a substantial fraction of PHF13 targets overlapped with unmethylated CpG islands at both active and Polycomb repressed promoters. It is noteworthy to mention that many of the factors mediating Polycomb recruitment to unmethylated CpG islands remain elusive. Recently it has been demonstrated that Pcl3, a protein that is important for stem cell self-renewal, is capable of recruiting PRC2 to CpG islands at a subset of targets via an interaction with SUZ12 (*Hunkapiller et al., 2012*). The co-occurrence of PHF13 and Polycomb in genes that were de-repressed after PHF13 depletion, and the fact that PHF13 interacts with PRC2 (*Figure 4*) make it tempting to speculate that PHF13 may as well be involved in the recruitment of chromatin effector complexes to specific H3K4me2/3 CpG enriched regions. Consistent with such a possibility, PHF13

depletion reduced the affiliation of SUZ12 and RNA PolII S5P with both H3K4me3 and H3K27me3 arguing for an integral involvement of PHF13 in these interactions.

The comparative analysis of the ChIPseq and RNAseq experiments (*Figure 6B and C*) indicates that PHF13 binds and regulates targets involved in transcription, cell cycle, chromosome organization, development and DNA binding. A role for PHF13 in differentiation, higher chromatin order and cell cycle has been previously described (*Bördlein et al., 2011*; *Kinkley et al., 2009*; *Mund et al., 2012*). What is newly identified is a role of PHF13 in the regulation of gene expression or transcription. Consistent with this possibility, several H3K4me2/3 readers have been previously reported to either positively or negatively impact transcription (*Fortschegger and Shiekhattar, 2011*; *Shi et al., 2006*). Perhaps the strongest line of support in favor of a transcriptional co-regulatory function is PHF13's affiliation with RNA polymerase II. PHF13 showed a strong correlation with promoter associated RNA polymerase II (namely hypophosphorylated, serine 5 phosphorylated and serine 7 phosphorylated) consistent with its specificity for H3K4me3. Nevertheless the strongest relationship was observed with RNA PolII S5P which is found at both active and polycomb repressed promoters (*Figure 5B*). RNA PolII S5P regulates efficient transcription and enhances splicing (*Hsin et al., 2014*), implicating PHF13 in these chromatin functions. Consistent with this possibility, we observed changes in gene expression upon PHF13 depletion (*Figure 6B and C* and *Figure 6—figure supplement 1*) as well as interactions with splicing factors by mass spectrometry analysis of PHF13 chromatin interaction partners (*Figure 4A* and *Figure 4—figure supplement 1* and *2*). While the interaction between PHF13 and splicing factors was not confirmed, based on these collective observations it warrants future efforts. Another interesting parallel is that RNA PolII S5P interacts with the H3K27me3 methyltransferase complex PRC2 (*Brookes et al., 2012*), as well as with H3K4me3 (Set1A, Set1B and MLL) methyltransferases (*Hsin and Manley, 2012*; *Hughes et al., 2004*; *Skalnikova et al., 2008*). Since the depletion of PHF13 leads to both a reduction of SUZ12 and RNA PolII S5P association with H3K27me3 and H3K4me3, it is possible that PHF13 also associates with H3K4me3 methyltransferases at active promoters in RNA PolII complexes similar to its affiliation with PRC2 and RNA PolII at repressed promoters. While this is currently just speculation, it is an interesting hypothesis and based on the current observations deserves future investigation.

A model depicting PHF13's ability to affiliate H3K4me3, DNA, PRC2 and RNA PolII to co-regulate gene expression is shown in *Figure 7C*. In this model we proposed that direct chromatin interactions with H3K4me3 and CpG rich DNA as well as indirect chromatin interactions with RNA PolII S5P (potentially also hypomethylated and S7 phosphorylated RNA PolII) and as of yet unknown transcriptional activating complexes, leads to their chromatin recruitment and or stabilization at active promoters to promote transcriptional activity. Likewise, direct chromatin interactions of PHF13 with H3K4me2/3 and CpG rich DNA and indirect chromatin interactions facilitated through PRC2 and RNA PolII S5P complexes leads to the recruitment and or stabilization of repressive PRC2 transcriptional complex at bivalent promoters, and thereby co-regulates gene repression.

Together our results demonstrate that PHF13 can contact chromatin in a multivalent fashion via direct interactions with DNA and H3K4me2/3 and indirectly by affiliation with Polycomb and RNA PolII complexes. ChIP sequencing revealed that in vivo PHF13 associates with distinct chromatin landscapes and that it shows a substantial overlap with H3K4me2/3, CpG rich DNA and hypo-, Ser5, and Ser7- phosphorylated RNA PolII, and a modest overlap with a subset of bivalent PRC2 targets. Furthermore, its depletion lead to a reduced interaction of RNA PolII S5P and SUZ12 with H3K4me3 and H3K27me3 and with each other, supporting that there are common complexes between these proteins and that PHF13 may act as a scaffold or bridging factor. Finally, PHF13 depletion in mESCs lead to both the up and down regulation of a subset of target genes functionally enriched in transcription, cell cycle, chromatin organization, DNA binding and developmental processes, explaining in part its ability to modulate such processes. Taken, together these data support that PHF13 is an H3K4me2/3 molecular reader and transcriptional co-regulator of H3K4me3 active and H3K4me3/H3K27me3 bivalent promoters.

## Materials and methods

### Cell lines

HeLa cells obtained from ATCC were used for GST-pull down experiments and were grown in DMEM supplemented with 1x Penicillin/Streptomycin, 10% FBS (Biochrome), 1x hepes and 1x sodium pyruvate. All other experiments were performed with mouse embryonic stem cells (mESCs) grown in Glassgow MEM (Sigma G5154), 20% FBS Hyclone – ThermoScientific SV30160.03, 1x Glutamax, 1x non-essential amino acids, 1x sodium pyruvate and 1.2 ml β-mercaptomethanol (Gibco 31350010). E14 mESCs were obtained originally from ATCC and were given to us by Prof. Luciano DiCroce. E14 mESCs were grown on gelatin coated plates. Inducible PHF13 shRNA mESCs are primary mESCs isolated from mice blastocytes and were purchased from Artemis Pharmaceuticals Gmbh. They were grown similar to E14 mESCs except that they were not grown on gelatin coated plates. Early passages required feeder cells but this was gradually eliminated by 10–15 passages. Both mESCs cell cultures were periodically stained for alkaline phosphatase activity (Millipore SCR004) to ensure they were significantly enriched (>90%) in stem cells and contained limited differentiated cells. All cell lines were routinely tested to ensure the absence of mycoplasma (Thermofisher Scientific C7028).

### Recombinant protein purification, biotin and GST-pull down and immunoprecipitation

#### Recombinant protein preparation

Purified GST proteins were generated using standard procedures. In brief, BL-21 bacteria carrying the GST expression vector were grown in 1x LB medium plus ampicillin overnight at 37°C under rotation (180 rpm). The next day the GST protein expression was induced using 1 mM IPTG and incubated for 4 hr at 30°C under rotation. The bacteria were then pelleted and stored at -80°C overnight. The pellet was then lysed in 10 ml of MTTBS buffer (50 mM Tris-HCl, pH 8.0, 150 mM NaCl, 5 mM EDTA pH 8.0 and 1% Triton X-100) supplemented with 30 mg lysozyme, 100 µl of RNAse A (10 mg/ml), 100 µl of DNAse I (50 µg/ml), 100 µl of complete protease inhibitor (Roche) and 100 µl of PMSF, for 10 min on ice. The bacterial lysates were then sonicated at high frequency for 3 x 15 s on ice, centrifuged at 13,000 rpm for 20 min at 4°C and the supernatant added to Gluthione agarose (500 µl) for 3h under rotation at 4°C. The beads were then washed with MTTBS (1x) and with PBS (2x) and frozen at −80°C until needed.

#### Biotinylated histone peptide pull downs

GST proteins were eluted from the beads in 250 µl of 40 mM reduced Glutathione for 2 hr at 4°C. The protein concentration of the elution was calculated. 1 µg of GST protein was incubated with 1 µg of biotinylated peptide and 10 µl Streptavidin Dynabeads (Life Sciences) in 300 µl of binding buffer (50 mM Tris- HCl pH 7.5, 0.1% NP40 and 300 mM NaCl) overnight at 4°C under rotation. The beads were then washed 5x with binding buffer and resuspended in 2x Laemmli buffer. The interactions were evaluated by SDS-PAGE and detected using a GST specific antibody (Invitrogen 71–7500).

#### GST Pull downs

Nuclei were prepared from HeLa in a hypotonic buffer (10 mM Tris HCl pH 7.4, 10 mM KCl and 15 mM $MgCl_2$) on ice for 10 min and then lysed with a CSK buffer (10 mM PIPES, 100 mM NaCl, 300 mM sucrose, 3 mM $MgCl_2$, 0.1% NP40, 1x complete protease inhibitor cocktail (Roche), 1 mM PMSF) for 15 min on ice. The chromatin was then pelleted by centrifugation and the supernatant (cytoplasm/nucleoplasm) was discarded. The chromatin was then resuspended in a nuclear lysis buffer (300 mM NaCl, 50 mM Hepes pH 7.5, 0.5%Igpal, 1 unit Benzonase (Merck), 2.5 mM $MgCl_2$,1 mM PMSF, 1x complete protease inhibitor cocktail (Roche), 1 mM $Na_3VO_4$ and1 mM NaF) for 30 min on ice. The debris was pelleted and the supernatant collected. The lysate was then diluted 1:10 for GST-pull downs with dilution buffer (50 mM HEPES pH 7.5, 5 mM EDTA and 0.05%IGEPAL) and incubated with 3 µg of GST proteins (still on the Glutathione agarose). The lysate bead mixture was incubated at 4°C for 1.5 hr under rotation and then washed 5x with PBS and resuspended in 2x

Laemmli buffer. The interactions were evaluated by SDS-PAGE using specific antibodies against H3K4me3 (Millipore 04–745).

## Immunoprecipitation

Immunoprecipitations were performed on nuclease digested lysates from E14 mESC cells using 20 ul of protein A agarose beads (Diagenode) containing 2 µg of either rabbit control IgG immunoglobulins (Diagenode - C15410206), rb pAb SUZ12 (Abcam – ab12073), rb pAb RNA PolIIS2P (Abcam – ab5095), rb pAb RNA PolII S5P (Abcam – ab5131) or rb pAb PHF13 specific antibody CR56. 1x 15cm dish of E14 mESC cells was used per IP and lysed first in a CSK buffer (10 mM PIPES, 100 mM NaCl, 300 mM sucrose, 3 mM $MgCl_2$, 0.1% NP40, 1x complete protease inhibitor cocktail (Roche), 1 mM PMSF) for 10 min on ice to isolate the cytoplasm. After centrifugation at 10,000 x G for 5 min, the supernatant was discarded and the pellet resuspended in a mild chromatin buffer (20 mM Tris-HCl pH 7.5, 100 mM NaCl, 0.5% Triton X, 2.5 mM $MgCl_2$, 2 mM $CaCl_2$) supplemented with 1x complete protease inhibitor cocktail (Roche) for 10 min on ice to isolate the nucleoplasm. After centrifugation at 10,000 x G for 5 min, the supernatant was discarded and the pellet resuspended in nuclease digestion buffer (50 mM Tris-HCl pH7.5, 100 mM NaCl and 2.5 mM $MgCl_2$ supplemented with 1x complete protease inhibitor cocktail (Roche) and 1 µl of Benzonase (Novagene – 70746–3)) for 10 min at RT and then for 30 min on ice to isolate the chromatin fraction of cellular lysate. The lysate was then centrifuged at 14,000 x G for 5 min and the supernatant collected. IPs were performed overnight at 4°C under rotation and then washed 5x in IP buffer, transferred to a new tube and denatured in Laemmli buffer. IPs were analyzed by immunoblot (4–15% gradient gels – Biorad) for precipitation of rt mAb PolIIS2 (Active Motif - 61083), mo mAb PolIIS5 (Abcam ab5408), rt mAb PolIIS7 (Active Motif - 61087), mo mAb SUZ12 (Millipore – 04–046), mo mAb EZH2 (Cell Signalling – 3147), rt mAb PHF13 (6F6- Generated by Elisabeth Kremmer- Helmholtz zentrum, Munich), rb mAb H3K4me3 (Millipore 04–745), rb pAb H3K27me3 (Diagenode – pAb-195-050).

## EMSA and nucleosome shift assays

DNA EMSA reactions were performed using a 248 bp DNA fragment radio-labeled by PCR, as previously described (*Hartlepp et al., 2005*) or a 40 bp Cy5-end labeled fragment (5'-Cy5-CCTGGAGAATCCCGGTGCCGAGGCCGCTCAATTGGTCGTA-3'; Eurofins MWG). The Cy5-labeled DNA was incubated with recombinant proteins in 1X binding buffer (50 mM Hepes pH7.6, 50 mM NaCl, 2 mM $MgCl_2$, 10% glycerol and 100 ng/µl BSA) for 20 min on ice and was analyzed by native PAGE. The label was detected with a Fujifilm Phosphoimager FLA-3000. Mononucleosomes for EMSAs were generated by salt gradient dialysis as previously described (*Dyer et al., 2004*) using recombinant Drosophila histone octamers and either a 200 bp DNA fragment or a 151 bp DNA fragment, both comprising derivatives of the Widom 601 nucleosome positioning sequence. The DNA fragments were released by restriction enzyme digest with NotI and XmaI, respectively, from plasmids comprising 12 repeats of the 200 bp fragment (*Huynh et al., 2005*) or four repeats of the 151 bp fragment (*Mueller-Planitz et al., 2013*). The XmaI ends of the 151 bp fragment were labeled with dCTP-Cy5 and Klenow polymerase. The mononucleosomes EMSA reactions were performed and analyzed like DNA EMSAs as described above. Unlabeled mononucleosomes were visualized by ethidium bromide staining.

## PHD domain purification, fluorescence polarization, ITC and crystallography

### Protein purification

Two fragments of human PHF13 (residues 232–281 and 226–280) covering the PHD domain were subcloned into a pET-28a-MHL vector via ligase-independent cloning. The recombinant proteins were over expressed in Escherichia coli BL21 (DE3) with the pRARE plasmid for codon-biased expression. Cells were grown in minimal media at 37°C to an optical density of approximately 2.5. Protein expression was induced with 1 mM isopropyl-1-thio-D-galactopuranoside and the cell cultures were grown for approximately 16 hr at 15°C after induction. The proteins were purified by affinity chromatography on Ni-NTA resin (Qiagen Mississauga, ON) and size exclusion chromatography using a Superdex 75 column (GE Healthcare, Tyrone, PA) in 20 mM Tris-HCl pH 8.0, 200 mM NaCl, 1 mM DTT and 20 µM $ZnCl_2$. The his-tag was cleaved from PHF13 (residues 226–280) by the

addition of 0.05 mg of TEV protease per milligram of PHF13 protein, followed by incubation in ice for 12 hr. The sample was then passed through a Ni-NTA column and the flow-through was collected and concentrated for crystallization. In another PHF13 construct (residues 232–281), the his-tag was not removed.

## Fluorescence polarization assays

The Fluorescence Polarization Assays were performed as described (*Xu et al., 2010*). All peptides used for fluorescence polarization and ITC measurements were synthesized by Tufts University Core Services (Boston, MA, USA). Binding assays were performed in 10 µL at a constant fluorescence labeled-peptide concentration of 40 nM and increasing amounts of PHF13 PHD domain at concentrations ranging from low to high micromolar in 20 mM Tris-HCl, pH 7.5, 150 mM NaCl, and 0.01% Tween-20. Fluorescence polarization assays were performed in 384 well plates, using a Synergy 2 microplate reader (BioTek, Vermont, USA). An excitation wavelength of 485 nm and an emission wavelength of 528 nm were used. The data were corrected for background of the free-labeled peptides. To determine the $K_D$ values, the data were fit to a hyperbolic function using Sigma Plot software (Systat Software, Inc., CA, USA).

## Isothermal titration calorimetry

Isothermal titration calorimetry measurements were recorded as described previously (*Xu et al., 2011*) at 25°C using a VP-ITC microcalorimeter (MicroCal Inc.). Experiments were performed by injecting 10 µl of peptide solution (2 mM) into a sample cell containing 100–200 µM of PHF13's PHD domain (226–280) in 20 mM Tris-HCl, pH 7.5, 150 mM NaCl. Different peptides (H3K4me3 and R2admK4me3) were dissolved and dialyzed into the same buffer as that of the PHF13's PHD domain. The concentrations of proteins and DNAs are estimated with absorbance spectroscopy using the extinction coefficient, $OD_{280}$ and $OD_{260}$, respectively. A total of 25–27 injections were performed with a spacing of 180 s and a reference power of 15 µcal s$^{-1}$. Binding isotherms were plotted and analyzed using Origin Software (MicroCal Inc.). The ITC measurements were fit to a one-site binding model.

## Protein crystallization

Purified PHF13 (residue 226–280) was mixed with methylated histone peptides by addition of a two-fold molar excess of peptide to the protein solution and crystallized at 18°C in 0.1M Tris-HCl pH 7.5, 1.5 M Na-Citrate, PEG400 6%. Crystals of PHF13 (residue 232–281) were obtained at 18°C in of 0.1M CHES pH 9.5 and 20% PEG8000.

## Structure determination

Diffraction experiments were performed at 100 K. Further experimental details are listed in *Figure 3—source data 1*. Diffraction intensities were indexed and scaled with HKL2000/HKL3000 (*Minor et al., 2006*). The structure of PHF13's PHD domain (232–281) was solved by the single wavelength anomalous diffraction method (*Wang, 1985*) using of the anomalous scattering from bound zinc ions and data collected on APS beam line SBC-CAT 19ID. Heavy atom substructure determination and initial phase calculation were carried out using SHELXD (*Schneider and Sheldrick, 2002*) and SHELXE (*Thorn and Sheldrick, 2013*), respectively. Automated model building in ARP/warp (*Perrakis et al., 2001*) was used to generate the initial model. Manual model improvements and refinement were carried out with COOT (*Emsley and Cowtan, 2004*) and REFMAC (*Vagin et al., 2004*), respectively. The MOLPROBITY server (*Davis et al., 2004*) was used for the validation of protein model geometry. The structure of PHF13-H3K4me3 was solved by molecular replacement using the program MOLREP (*Vagin and Teplyakov, 2000*).

## Chromatography

Chromatography was performed using nuclease digested chromatin lysates obtained from 14 (15cm) plates of E14 ES cells and run on a Superose 6 10/300 GL column (GE Healthcare) in a buffer composed of 50 mM Tris-HCl (pH 7.5) and 100 mM NaCl with a flow rate of 0.3 ml/min. Fractions were collected every 2 min.

### LC-MS/MS

#### Trypsin Digested PHF13 IPs

The HiPPR™ Detergent Removal Resin Kit (Thermo Scientific, Waltham, MA) was used to remove remaining Triton X contaminations in magnetic beads samples, according to manufactures instructions. Proteins were subsequently digested with 100 ng trypsin (Roche) at pH 8.0 on a rocking platform at 37°C, overnight. Tryptic digests were acidified by 1% final concentration of formic acid, cleaned by C18 StageTips (Thermo Scientific, Waltham, MA) and lyophilized.

#### LysC Digested PHF13 IPs

PHF13 and control IPs were run into 4–15% gradient gels (Biorad) for 12 min at 100V and then excised from the gel. Gel samples were reduced in 50 mM DTT at 56°C for 1 hr and alkylated with a final concentration of 5.5 mM chloroacetamide for 30 min. In gel digestion was done by 1 μg LysC (Roche) incubation at 24°C for 24 hr, according to (*Kaiser et al., 2008*).

#### All samples

Each sample was dissolved in 5% acetonitrile and 2% formic acid prior injection onto a nanoflow reverse-phase liquid chromatography (Dionex Ultimate 3000, Thermo Scientific) coupled online to a Thermo Scientific Q-Exactive HF Orbitrap mass spectrometer (nanoLC-MS/MS). Briefly, the LC separation was performed using a PicoFrit analytical column (75 μm ID × 55 cm long, 15 μm Tip ID (New Objectives, Woburn, MA) packed in-house with 3 μm C18 resin (Reprosil-AQ Pur, Dr. Maisch, Ammerbuch-Entringen, Germany) under 50°C controlled temperature. Peptides were eluted using a gradient from 3.8 to 40% solvent B in solvent A over 60 min at 266 nL per minute flow rate. Solvent A was 0.1% formic acid and solvent B was 79.9% acetonitrile, 20% $H_2O$, 0.1% formic acid). Nanoelectrospray was generated by applying 3.5 kV. A cycle of one full Fourier transformation scan mass spectrum (300$-$1750 m/z, resolution of 60,000 at m/z 200) was followed by 12 data-dependent MS/MS scans with a normalized collision energy of 25 eV. MS data were analyzed by MaxQuant (v1.5.3.3) (*Cox and Mann, 2008*), and searched against the murine (Mus musculus) proteome database UniProtKB with 52.490 entries, released in 11/2014, using a FDR of 0.01 for proteins and peptides with a minimum length of 7 amino acids. A maximum of five missed cleavages in the tryptic gel digests and two in all other samples were allowed. Cysteine carbamidomethylation was set as a fixed modification accordingly, while N-terminal acetylation and methionine oxidation were set as variable modifications. Only proteins with at least two unique peptides in IP samples and no entries in controls were regarded as interaction partners, otherwise manually verified $MS^2$ spectra were listed in *Figure 4—figure supplement 1*.

### Chromatin immunoprecipitation

Chromatin immunoprecipitation was performed using a standard technique. In brief E14 ES cells were grown in the presence of LIF and 20% FBS. The cells were collected and cross-linked for 15 min with 1% formaldehyde. The fixation reaction was quenched by the addition of glycine (0.125 M) for 10 min. The cells were then lysed on ice for 10 min in 1.3 ml of ChIP buffer, 1 volume of SDS buffer (100 mM NaCl, 50 mM Tris-HCl pH 8, 5 mM EDTA, 0.2% $NaN_3$ and 0.5% SDS): 0.5 volume of Triton dilution buffer (100 mM Tris-HCl pH8.6, 100 mM NaCl, 5 mM EDTA, 0.2% $NaN_3$ and 5% Triton-X 100). The lysate was passed through a syringe multiple times and then sonicated in Bioruptor at high frequency for 2 x 6 cycles. The lysate was then centrifuged at 13,000 rpm for 30 min to pellet debris and a protein determination was measured. 1 mg of chromatin was then incubated with 5 μl of PHF13 antibody (CR56) or rabbit IgG control antibody overnight at 4°C under rotation. 30 μl of protein A beads were then added to the reaction and allowed to incubate at 4°C for 2h. The beads were then washed by successive low salt (0.1% SDS, 1% Triton X, 2 mM EDTA, 20 mM Tris-HCl and 150 mM NaCl), high salt (0.1% SDS, 1% Triton X, 2 mM EDTA, 20 mM Tris-HCl and 500 mM NaCl) and LiCl washes and then eluted for 3h at 65°C with 110 μl of elution buffer (1% SDS and 100 mM $NaHCO_3$). The RNA and proteins were then digested with RNAse A and Proteinase K and the DNA purified and sequenced.

## Stable and Doxycyclin inducible Phf13 shRNA mESC cell lines

### Generation of stable shRNA E14 cell lines

Mission-pLKO.1 lenti-viruses (Sigma Aldrich) for Phf13 shRNA (Mission Phf13 shRNA3 TRCN0000241824) and control scrambled shRNA (Mission ShControl– SHC001) were generated by calcium phosphate co-transfection of 18 ng of DNA (8.8 μg pLKO, 7.3 μg Δ891 and 1.8 μg VSV-g) in 15 cm dishes of 293T cells. The next day the cells were washed and the medium was replaced and 48 hr post transfection, the supernatants containing lenti-viruses were collected. 6 ml of lenti-viral medium supplemented with 20% FCS and 20 μl β-mercaptomethanol (Gibco 31350010) was diluted with 3ml fresh medium, 9 μl Pb, 20 μl LIF and 1 ml of E14 mESCs (3 x $10^6$ cells), mixed in a conical tube and plated on 10 cm gelatin coated plates.The next day medium was replaced with fresh E14 mESC medium and the cells were put under Puromyocin selection (4 μg/ml). The cells were split every two days and kept under selection. After 12 days the cells were harvested for RNA sequencing. The shRNA sequence for Mission PHF13 shRNA3 TRCN0000241824 is CCGGCCGGGAC TCCAAGTTTGATATCTCGAGATATCAAACTTGGAGTCCCGGTTTTTG.

Doxycyclin Inducible Phf13 ShRNA mESCs: Inducible Phf13 RNAi primary mESCs were generated by Artemis Pharmaceuticals Gmbh, using site specific recombination approach (RMCE exchange) into the Rosa26 locus. The RNAi sequence that was used is GCACCTTCGTCTTGGCATATG.

## RNAseq

Total cellular RNA was collected from wild-type E14 ES cells or from PHF13 depleted ES cells using RNeasy plus mini-kit (Qiagen) according to manufactures instructions. 1.5 μg of total RNA was then depleted of ribosomal RNA using the Ribo zero magnetic kit (Epicenter) according to the manufacturer's directions. The rRNA depleted RNA was then measured by Qubit and 20 ng was used to generate the RNA library using ScriptSeq v2 RNA-Seq library preparation kit (Epicenter) according to the manufacturer's instructions.

## Bioinformatic analysis

### RNAseq data

The paired-end 50 reads were aligned to the mouse genome (downloaded from ENSEMBL: Mus_musculus.GRCm38.dna.primary_assembly.fa) using STAR (version 2.5.1b_modified; PMID: 23104886). The index was built with the –sjdbGTFfile option set to "Mus_musculus.GRCm38.83.gtf".

Differential gene expression was determined by DESeq2 (PMID: 25516281). Counting was performed using the Bioconductor package bamsignals.

### ChIP-seq data

The ChIP-seq reads (*Marks et al., 2009*; *Marks et al., 2012*; *Morey et al., 2013*; *Ng et al., 2013*; *Xiao et al., 2012*) for each experiment were individually mapped against the mouse genome (genome version mm10) using Bowtie2 with default parameters (*Langmead and Salzberg, 2012*). The resulting SAM file was converted into BAM, sorted and indexed using samtools (*Li et al., 2009*). BigWig files were generated from the individual, unmerged BAM files using igvtools count (*Thorvaldsdottir et al., 2013*) with the parameters: –minMapQuality 30 –postExtFactor 200, whose output was piped into wigToBigWig (*Kuhn et al., 2013*) using the parameters: -clip. After visual inspection, we removed those experiments which where markedly different from the rest. The remaining BAM files corresponding to ChIP-seq experiments with the same target were merged; alignments with a mapping quality less than 30 and unmapped reads were discarded using samtools. Corresponding BigWig files were merged using bigWigMerge (*Kuhn et al., 2013*).

The DNase HS reads were individually mapped against the mouse genome (genome version mm10) using Bowtie2 with default parameters. The resulting SAM file was converted into BAM, sorted and indexed using samtools. BigWig files were generated from the individual, unmerged BAM files using igvtools count with the parameters: –minMapQuality 30 –postExtFactor 1, whose output was piped into wigToBigWig using the parameters: -clip

## To call ChIP-seq peaks MACS2 was used with the following settings

–treatment <BAM file> –control <BAM file> –format BAM –gsize mm –nomodel –shift-size 100 – keep-dup auto –call-summits (*Zhang et al., 2008*). In case of the PHF13 ChIPseq data we used a matched IgG control, while for the other ChIPseq targets we used a merged Input control.

## To call DHS we used also MACS2 with the following settings

–treatment <BAM file> –format BAM –gsize mm –nomodel –shiftsize 100 –keep-dup 1 –call-summits.

The Venn diagrams were generated using the R-package DiffBind and VennDiagram (*Ross-Innes et al., 2012*). All PHF13 peaks were intersected with the corresponding peaks of the other ChIPseq targets.

PHF13 peak centered heatmaps were generated with an in-house developed script written in R. For each ChIPseq target the read counts (in non-overlapping 51 base pair bins) were obtained in a window spanning +/- 5100 base pairs around PHF13 peak summits inferred by MACS2. Counting was performed directly from the BAM files using the Bioconductor package bamsignals. For all ChIP-seq tracks the 5' ends were shifted 100 base pairs downstream, while the DHS track was left unshifted. The resulting matrices were further processed by identifying the value corresponding to the 99.9% quantile of read counts, setting all matrix entries larger than this value to this value and dividing by the value. The so-obtained matrices except the one corresponding to PHF13 were concatenated and served as input to the k-means clustering algorithm with two cluster centers. The obtained clustering was used to order the PHF13 ChIPseq peaks first by cluster and second by the peak score reported by MACS2.

The boxplots were generated from control normalized average ChIPseq signals in a region +/- 1500 base pairs around the transcription start site (TSS). First, reads were counted using bamsignals for each of the TSSs. Second, we used an in house developed R package, normR, to normalized the data. Briefly, normR fits a binomial mixture model, where one component corresponds to the background and the other to signal. Using the background component, we estimate pseudo counts and add these to the observed counts (this leads to a regularization, where low read counts are automatically shifted to zero). If $r_i$ is the number of reads in the control and $s_i$ the number of reads from the ChIP counted at TSS $i$, we calculate the following quantity:

$$e_i^* = \ln\left(\frac{s_i + \alpha_s}{r_i + \alpha_r} \times \frac{\alpha_r}{\alpha_s}\right)$$

where $\alpha_s$ and $\alpha_r$ are the pseudo counts. Finally, we 'normalize' by the average log enrichment $\ln\langle f \rangle$ obtained in the signal component

$$e_i = \frac{e_i^*}{\ln\langle f \rangle}.$$

For each gene we determined the promoter with the highest H3K4me3 enrichment. Shown in the boxplots is $e_i$ *for those promoters* as estimate for the signal strength grouped by either down- or up-regulated or unchanged genes upon PHF13 knockdown with an adjusted p-value cutoff of 0.05. The boxplots were generated using the R function boxplot.

### GO analysis

ChIPSeq targets of PHF13, RNASeq up or down-regulated genes were collected and served as input to DAVID/EASE. Overrepresented GO terms for the ontologies Biological Process (BP_FAT), Molecular Function (MF_FAT), and Cellular Component (CC_FAT) were downloaded and visualized with the R-package wordcloud, where the -10 * $\log_{10}$(adj. p-value) was used to determine the word size.

## ChIPseq and DNase hypersensitivity data:

Accession numbers are from the short read archive (http://www.ncbi.nlm.nih.gov/sra).

| ChIPseq Target | Accession number | Reference |
| --- | --- | --- |
| H3K4me1 | SRX122631 | *Xiao et al., 2012* |
| H3K4me1 | SRX185842 | Mouse ENCODE |
| H3K4me1 | SRX186789 | Mouse ENCODE |
| H3K4me2 | SRR414940 | *Xiao et al., 2012* |
| H3K4me3 | SRX026272 | *Marks et al., 2012* |
| H3K4me3 | SRX149184 | *Ng et al., 2013* |
| H3K4me3 | SRX149188 | *Ng et al., 2013* |
| H3K4me3 | SRX185845 | Mouse ENCODE |
| H3K4me3 | SRX186795 | Mouse ENCODE |
| H3K27me3 | SRX006976 | *Marks et al., 2012* |
| H3K27me3 | SRX026276 | *Marks et al., 2012* |
| H3K27me3 | SRX122629 | *Xiao et al., 2012* |
| EZH2 | SRR501769 | *Marks et al., 2012* |
| SUZ12 | SRX206420 | *Morey et al., 2013* |
| DHS | SRX191012 | Mouse ENCODE |
| H3K9me3 | SRX112916 | *Marks et al., 2012* |
| H3K9me3 | SRX186790 | Mouse ENCODE |
| PolII-8WG16 | SRR391036, SRR391037 | *Brookes et al., 2012* |
| PolII-S2P | SRR391034, SRR391038,SRR391039 | *Brookes et al., 2012* |
| PolII-S5P | SRR391032, SRR391033, SRR391050 | *Brookes et al., 2012* |
| PolII-S7P | SRR391035, SRR391040 | *Brookes et al., 2012* |

## Acknowledgements

We would like to acknowledge Silke Sperling for fruitful discussions and for contributing biotinylated histone peptides, Dr. Robert Lam for his assistance in data collection and structure determination of PHF13, Beate Lukaszewsa-McGreal for her excellent technical support in LC-MS/MS sample preparation and the sequencing core unit at the Max Planck for Molecular genetics for their excellent technical support in the RNA sequencing experiments. The structural data have been deposited in the Protein Data Bank with the accession numbers 3O70 and 3O7A for apo-PHF13-PHD and H3K4me3-PHF13-PHD, respectively. The ChIP and RNA sequencing data have been deposited to Array Express (https://www.ebi.ac.uk/arrayexpress/) and can be accessed with the following ids E-MTAB-2636 and E-MTAB-2637, respectively. This work was mainly funded by a grant from the Deutsche Krebshilfe #10-2243-Wi4 and by the Max Planck Society. The Heinrich Pette Institute is supported by the Freie und Hansestadt Hamburg and the German Federal Ministry of Health and Social Security. The SGC is a registered charity (number 1097737) that receives funds from AbbVie, Bayer Pharma AG, Boehringer Ingelheim, Canada Foundation for Innovation, Eshelman Institute for Innovation, Genome Canada, Innovative Medicines Initiative (EU/EFPIA) [ULTRA-DD grant no. 115766], Janssen, Merck & Co., Novartis Pharma AG, Ontario Ministry of Economic Development and Innovation, Pfizer, São Paulo Research Foundation-FAPESP, Takeda, and the Wellcome Trust (to JM).

## Additional information

### Funding

| Funder | Grant reference number | Author |
| --- | --- | --- |
| Deutsche Krebshilfe | 10-2243-Wi4 | Hans Will |

The funders had no role in study design, data collection and interpretation, or the decision to submit the work for publication.

## Author contributions

H-RC, Performed all bioinformatic analysis of the manuscript, Involved in the writing of the manuscript, Acquisition of data, Analysis and interpretation of data; CX, CB, Performed the fluorescence polarization, isothermal titration calorimetry and crystallography in the lab of JM, Acquisition of data, Analysis and interpretation of data; AF, Performed RNA sequencing and qPCR analysis in lab of HC, Acquisition of data, Analysis and interpretation of data, Drafting or revising the article; AM, Performed EMSA and nucleosome shift assays, Acquisition of data, Analysis and interpretation of data; ML, Performed ChIP sequencing in the lab of LD, Acquisition of data, Analysis and interpretation of data; HS, Involved in the conceptual design of the manuscript and contributed critical information and materials, Analysis and interpretation of data; TS, Conception and design, Contributed unpublished essential data or reagents; ID, Performed RNA sequencing and qPCR analysis in lab of HC, Acquisition of data, Analysis and interpretation of data; AE, Performed EMSA and nucleosome shift assays, Provided materials and expertise for the EMSA and nucleosome shift assays, Acquisition of data; CR, HK, Provided materials and expertise for the EMSA and nucleosome shift assays, Analysis and interpretation of data; DM, Performed the mass spectrometry analysis of PHF13 interacting proteins, Acquisition of data, Analysis and interpretation of data; LC, Performed sequencing and raw analysis of the ChIP sequencing, Acquisition of data, Analysis and interpretation of data; AW, Involved in the conceptual design of the manuscript and contributed critical information and materials; LDC, Conception and design, Analysis and interpretation of data, Contributed unpublished essential data or reagents; JM, Involved in the writing of the manuscript, Conception and design, Analysis and interpretation of data; HW, Involved in the conceptual design of the manuscript and contributed critical information and materials, Analysis and interpretation of data, Drafting or revising the article; SK, Performed the biochemical experiments in the labs of HW and HC, Performed the ChIP sequencing in the lab of LD, Performed the EMSA and Nucleosome shift assays, Acquisition of data, Analysis and interpretation of data, Involved in the conceptual design of the manuscript and wrote the manuscript

## Author ORCIDs

Chao Xu, http://orcid.org/0000-0003-0444-7080
Martin Lange, http://orcid.org/0000-0003-1650-533X
Luca Cozzuto, http://orcid.org/0000-0003-3194-8892
Sarah Kinkley, http://orcid.org/0000-0003-4997-4749

# Additional files

## Major datasets

The following datasets were generated:

| Author(s) | Year | Dataset title | Dataset URL | Database, license, and accessibility information |
|---|---|---|---|---|
| Ho-Ryun Chung, Sarah Kinkley | 2014 | PHF13 ChIP sequencing in mESCs | https://www.ebi.ac.uk/arrayexpress/experiments/E-MTAB-2636/ | Publicly available at EBI ArrayExpress (accession no: E-MTAB-2636) |
| Lam R, Bian CB, Xu C, Kania J, Bountra C, Weigelt J, Arrowsmith CH, Edwards AM, Bochkarev A, Min J, Structural Genomics Consortium | 2010 | PHD-type zinc finger of human PHD finger protein 13 | http://www.rcsb.org/pdb/explore/explore.do?structureId=3O70 | Publicly available at Protein Data Bank (accession no: PDB 3O70) |

| Bian CB, Lam R, Xu C, Bountra C, Arrowsmith CH, Weigelt J, Edwards AM, Bochkarev A, Min J | 2010 | Crystal structure of PHF13 in complex with H3K4me3 | http://www.rcsb.org/pdb/explore/explore.do?structureId=3O7A | Publicly available at Protein Data Bank (accession no: PBD 3O7A) |
| | 2014 | RNA sequencing in mESCs after PHF13 depletion | https://www.ebi.ac.uk/arrayexpress/experiments/E-MTAB-2637/ | Publicly available at EBI ArrayExpress (accession no: E-MTAB-2637) |

The following previously published datasets were used:

| Author(s) | Year | Dataset title | Dataset URL | Database, license, and accessibility information |
|---|---|---|---|---|
| Xiao S, Xie D, Cao X, Yu P, Xing X, Chen CC, Musselman M, Xie M, West FD, Lewin HA, Wang T, Zhong S | 2012 | GSM881352: E14_d0_H3K4me1_ChIP-seq | http://www.ncbi.nlm.nih.gov/sra/?term=SRX122631 | Publicly available at NCBI Sequence Read Archive (accession no: SRX122631) |
| Mouse ENCODE | 2015 | GSM1000121: LICR_ChipSeq_ES-E14_H3K4me1_E0 | http://www.ncbi.nlm.nih.gov/sra/?term=SRX185842 | Publicly available at NCBI Sequence Read Archive (accession no: SRX185842) |
| Mouse ENCODE | 2015 | GSM1003750: Stanford_ChipSeq_ES-E14_H3K4me1_std | http://www.ncbi.nlm.nih.gov/sra/?term=SRX186789 | Publicly available at NCBI Sequence Read Archive (accession no: SRX186789) |
| Xiao S, Xie D, Cao X, Yu P, Xing X, Chen CC, Musselman M, Xie M, West FD, Lewin HA, Wang T, Zhong S | 2012 | GSM881353: E14_d0_H3K4me2_ChIP-seq | http://www.ncbi.nlm.nih.gov/sra/?term=SRR414940 | Publicly available at NCBI Sequence Read Archive (accession no: SRR414940) |
| Marks H, Kalkan T, Menafra R, Denissov S, Jones K, Hofemeister H, Nichols J, Kranz A, Stewart AF, Smith A, Stunnenberg HG | 2012 | GSM590111: E14-serum_H3K4me3 | http://www.ncbi.nlm.nih.gov/sra/?term=SRX026272 | Publicly available at NCBI Sequence Read Archive (accession no: SRX026272) |
| Ng JH, Kumar V, Muratani M, Kraus P, Yeo JC, Yaw LP, Xue K, Lufkin T, Prabhakar S, Ng HH | 2013 | GSM936109: Replicate1-ES-WGA-H3K4me3 | http://www.ncbi.nlm.nih.gov/sra/?term=SRX149184 | Publicly available at NCBI Sequence Read Archive (accession no: SRX149184) |
| Ng JH, Kumar V, Muratani M, Kraus P, Yeo JC, Yaw LP, Xue K, Lufkin T, Prabhakar S, Ng HH | 2013 | GSM936113: ES-nonWGA-H3K4me3 | http://www.ncbi.nlm.nih.gov/sra/?term=SRX149188 | Publicly available at NCBI Sequence Read Archive (accession no: SRX149188) |
| Mouse ENCODE | 2015 | GSM1000124: LICR_ChipSeq_ES-E14_H3K4me3_E0 | http://www.ncbi.nlm.nih.gov/sra/?term=SRX185845 | Publicly available at NCBI Sequence Read Archive (accession no: SRX185845) |
| Mouse ENCODE | 2015 | GSM1003756: Stanford_ChipSeq_ES-E14_H3K4me3_std | http://www.ncbi.nlm.nih.gov/sra/?term=SRX186795 | Publicly available at NCBI Sequence Read Archive (accession no: SRX186795) |

| | | | | |
|---|---|---|---|---|
| Marks H, Kalkan T, Menafra R, Denissov S, Jones K, Hofemeister H, Nichols J, Kranz A, Stewart AF, Smith A, Stunnenberg HG | 2009 | H3K27me3_ChIP-Seq_E14_undiff | http://www.ncbi.nlm.nih.gov/sra/?term=SRX006976 | Publicly available at NCBI Sequence Read Archive (accession no: SRX006976) |
| Marks H, Kalkan T, Menafra R, Denissov S, Jones K, Hofemeister H, Nichols J, Kranz A, Stewart AF, Smith A, Stunnenberg HG | 2012 | GSM590115: E14-serum_H3K27me3 | http://www.ncbi.nlm.nih.gov/sra/?term=SRX026276 | Publicly available at NCBI Sequence Read Archive (accession no: SRX026276) |
| Xiao S, Xie D, Cao X, Yu P, Xing X, Chen CC, Musselman M, Xie M, West FD, Lewin HA, Wang T, Zhong S | 2012 | GSM881350: E14_d0_H3K27me3_ChIP-seq | http://www.ncbi.nlm.nih.gov/sra/?term=SRX122629 | Publicly available at NCBI Sequence Read Archive (accession no: SRX122629) |
| Morey L, Aloia L, Cozzuto L, Benitah SA, Di Croce L | 2013 | GSM1041373: Cbx7_ChIPSeq | http://www.ncbi.nlm.nih.gov/sra/?term=SRX206419 | Publicly available at NCBI Sequence Read Archive (accession no: SRX206419) |
| Morey L, Aloia L, Cozzuto L, Benitah SA, Di Croce L | 2013 | GSM1041372: Ring1B_ChIPSeq | http://www.ncbi.nlm.nih.gov/sra/?term=SRX206418 | Publicly available at NCBI Sequence Read Archive (accession no: SRX206418) |
| Marks H, Kalkan T, Menafra R, Denissov S, Jones K, Hofemeister H, Nichols J, Kranz A, Stewart AF, Smith A, Stunnenberg HG | 2012 | GSM590132: E14-serum_Ezh2 | http://www.ncbi.nlm.nih.gov/sra/?term=SRR501769 | Publicly available at NCBI Sequence Read Archive (accession no: SRR501769) |
| Morey L, Aloia L, Cozzuto L, Benitah SA, Di Croce L | 2013 | GSM1041374: Suz12_ChIPSeq_2 | http://www.ncbi.nlm.nih.gov/sra/?term=SRX206420 | Publicly available at NCBI Sequence Read Archive (accession no: SRX206420) |
| Mouse ENCODE | 2012 | GSM1014154: UW_DnaseSeq_ES-E14_E0_129/Ola | http://www.ncbi.nlm.nih.gov/sra/?term=SRX191012 | Publicly available at NCBI Sequence Read Archive (accession no: SRX191012) |
| Marks H, Kalkan T, Menafra R, Denissov S, Jones K, Hofemeister H, Nichols J, Kranz A, Stewart AF, Smith A, Stunnenberg HG | 2012 | GSM850406: E14-serum_H3K9me3 | http://www.ncbi.nlm.nih.gov/sra/?term=SRX112916 | Publicly available at NCBI Sequence Read Archive (accession no: SRX112916) |
| Mouse ENCODE | 2015 | GSM1003751: Stanford_ChipSeq_ES-E14_H3K9me3_std | http://www.ncbi.nlm.nih.gov/sra/?term=SRX186790 | Publicly available at NCBI Sequence Read Archive (accession no: SRX186790) |
| Brookes E, de Santiago I, Hebenstreit D, Morris KJ, Carroll T, Xie SQ, Stock JK, Heidemann M, Eick D, Nozaki N, Kimura H, Ragoussis J, Teichmann SA, Pombo A, Xie JK | 2012 | GSM850467: RNAPII S5P ChIPSeq | http://www.ncbi.nlm.nih.gov/sra/?term=SRR391032 | Publicly available at NCBI Sequence Read Archive (accession no: SRR391032) |

| | | | | |
|---|---|---|---|---|
| Brookes E, de Santiago I, Hebenstreit D, Morris KJ, Carroll T, Xie SQ, Stock JK, Heidemann M, Eick D, Nozaki N, Kimura H, Ragoussis J, Teichmann SA, Pombo A, Xie JK | 2012 | GSM850467: RNAPII S5P ChIPSeq | http://www.ncbi.nlm.nih.gov/sra/?term=SRR391033 | Publicly available at NCBI Sequence Read Archive (accession no: SRR391033) |
| Brookes E, de Santiago I, Hebenstreit D, Morris KJ, Carroll T, Xie SQ, Stock JK, Heidemann M, Eick D, Nozaki N, Kimura H, Ragoussis J, Teichmann SA, Pombo A, Xie JK | 2012 | GSM850470: RNAPII S2P ChIPSeq | http://www.ncbi.nlm.nih.gov/sra/?term=SRR391034 | Publicly available at NCBI Sequence Read Archive (accession no: SRR391034) |
| Brookes E, de Santiago I, Hebenstreit D, Morris KJ, Carroll T, Xie SQ, Stock JK, Heidemann M, Eick D, Nozaki N, Kimura H, Ragoussis J, Teichmann SA, Pombo A, Xie JK | 2012 | GSM850468: RNAPII S7P ChIPSeq | http://www.ncbi.nlm.nih.gov/sra/?term=SRR391035 | Publicly available at NCBI Sequence Read Archive (accession no: SRR391035) |
| Brookes E, de Santiago I, Hebenstreit D, Morris KJ, Carroll T, Xie SQ, Stock JK, Heidemann M, Eick D, Nozaki N, Kimura H, Ragoussis J, Teichmann SA, Pombo A, Xie JK | 2012 | GSM850469: RNAPII 8WG16 ChIPSeq | http://www.ncbi.nlm.nih.gov/sra/?term=SRR391036 | Publicly available at NCBI Sequence Read Archive (accession no: SRR391036) |
| Brookes E, de Santiago I, Hebenstreit D, Morris KJ, Carroll T, Xie SQ, Stock JK, Heidemann M, Eick D, Nozaki N, Kimura H, Ragoussis J, Teichmann SA, Pombo A, Xie JK | 2012 | GSM850469: RNAPII 8WG16 ChIPSeq | http://www.ncbi.nlm.nih.gov/sra/?term=SRR391037 | Publicly available at NCBI Sequence Read Archive (accession no: SRR391037) |
| Brookes E, de Santiago I, Hebenstreit D, Morris KJ, Carroll T, Xie SQ, Stock JK, Heidemann M, Eick D, Nozaki N, Kimura H, Ragoussis J, Teichmann SA, Pombo A, Xie JK | 2012 | GSM850470: RNAPII S2P ChIPSeq | http://www.ncbi.nlm.nih.gov/sra/?term=SRR391038 | Publicly available at NCBI Sequence Read Archive (accession no: SRR391038) |
| Brookes E, de Santiago I, Hebenstreit D, Morris KJ, Carroll T, Xie SQ, Stock JK, Heidemann M, Eick D, Nozaki N, Kimura H, Ragoussis J, Teichmann SA, Pombo A, Xie JK | 2012 | GSM850470: RNAPII S2P ChIPSeq | http://www.ncbi.nlm.nih.gov/sra/?term=SRR391039 | Publicly available at NCBI Sequence Read Archive (accession no: SRR391039) |

| | | | | |
|---|---|---|---|---|
| Brookes E, de Santiago I, Hebenstreit D, Morris KJ, Carroll T, Xie SQ, Stock JK, Heidemann M, Eick D, Nozaki N, Kimura H, Ragoussis J, Teichmann SA, Pombo A, Xie JK | 2012 | GSM850468: RNAPII S7P ChIPSeq | http://www.ncbi.nlm.nih.gov/sra/?term=SRR391040 | Publicly available at NCBI Sequence Read Archive (accession no: SRR391040) |
| Brookes E, de Santiago I, Hebenstreit D, Morris KJ, Carroll T, Xie SQ, Stock JK, Heidemann M, Eick D, Nozaki N, Kimura H, Ragoussis J, Teichmann SA, Pombo A, Xie JK | 2012 | GSM850475: RNAPII S5PRepeat ChIPSeq | http://www.ncbi.nlm.nih.gov/sra/?term=SRR391050 | Publicly available at NCBI Sequence Read Archive (accession no: SRR391050) |

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
