## [Decision Letter]

Thank you for submitting your work entitled "PHF13 is a Molecular Reader and Transcriptional Co-Regulator of H3K4me2/3" for peer review at *eLife*. Your submission has been overall favorably evaluated by James Manley as the Senior editor, a Reviewing editor, and three reviewers.

The reviewers have discussed the reviews with one another and the Reviewing editor has drafted this decision to help you prepare a revised submission.

Because the first two reviewers had quite opposing views on the significance of the findings and rigor of some of the science, we decided to have a third reviewer examine the paper. In general it was felt that the findings of Figure 1–Figure 4 were solid and were of interest, but the messages and experiments in Figure 5–Figure 7 would benefit from the following experiments to provide insight into the identity of the PHF13 containing complexes and their function. For publication in *eLife*:

Essential revisions:

1) You should carry out anti-PHF13 IPs from fractions 16-20, 22-26, and 27-30 (see Figure 5) and analyse the co-IP-ed complexes by mass spec to identify the subunits, which are stoichiometrically associating with PHF13 in the different fractions.

2) You need to examine if there is any 'free' PHF13 in sESCs and interpret the ChIPseq data accordingly? Without this knowledge, it is very difficult to understand why there are so few overlapping ChIP-seq binding sites between PHF13 and SUZ12 or between PHF13 and EZH2.

3) You need to state what type of RNA was analyzed by RNA-seq. If it was total mRNA, then you need to tone down the interpretations on transcriptional regulation, because the state of the field now is to look at nascent RNA.

4) Validation of the ChIP-seq results is required by ChIP-qRT PCR for example.

5) Demonstration that similar RNA changes were observed with more than one siRNA is necessary to prove these were not off target effects. This could also be done by qRT PCR.

6) Determine whether the interaction between the PHF13 PHD finger and histone H3K4 methylation is required for PHF13's function.

---

## [Author Response]

1) You should carry out anti-PHF13 IPs from fractions 16-20, 22-26, and 27-30 (see Figure 5) and analyse the co-IP-ed complexes by mass spec to identify the subunits, which are stoichiometrically associating with PHF13 in the different fractions.

We have replaced Figure 5 from the original manuscript with Figure 4 in the revised manuscript.

In the original manuscript we analyzed the size exclusion profile of PHF13 in U20S cells and observed a partial overlap of PRC2 with the elution profile of PHF13. Furthermore, we showed that PHF13 could co-immunoprecipitate the PRC2 complex members SUZ12, EZH2 and EED. Through the technique of FACS-FRET we determined that the interaction of PHF13 with PRC2 was likely via SUZ12 as no (“direct”- i.e. less than 10Å) interaction was observed using this approach with EZH2 or EED.

In the original manuscript we were unaware of the interaction between PHF13 and RNA PolII. This interaction was identified during this revision due to the request that we perform mass spectrometry from PHF13 immunoprecipitations to help shed some light on “the identification of the different PHF13-containing positive and negative co-regulatory complexes (as suggested in Figure 7)”.

In the revised manuscript, we have repeated the size exclusion chromatography experiments in E14 mESCs in a partial effort to address the concern that the anti-PHF13 IP WB, the FRET and the ChIP-seq data do not fit together. As the ChIP- and RNA-sequencing were performed in E14 mESCs we felt that we should address point 1 as well in E14 mESCs to make the data more consistent and comparable. The running buffer and elution rate needed to be altered, see Materials and methods, in mESCs to obtain ideal fractionation according to the elution profile of chromatography molecular standards.

As suggested we carried out anti-PHF13 IPs from high (13-17), mid (18-26) and lower (27-33) molecular weight fractions (see Figure 4). The fraction pooling, was based on the elution profiles of RNA PolII S2P, S5P and S7P, SUZ12, EZH2, PHF13, H3K4me3 and H3K27me3. The immunoblot indicated potentially 3 different PHF13 containing complexes (Figure 4). In fractions 14-16 PHF13 co-eluted with RNA PolII, H3K4me3 and H3K27me3. In fractions 18-26 PHF13 co-eluted with RNA PolII, PRC2 (EZH2 and SUZ12), H3K4me3 and H3K27me3, while in fractions 28-32 PHF13 co-eluted only with RNA PolII.

Immunoprecipitation of PHF13 from these different fractions indicated that PHF13, RNA PolII and PRC2 (SUZ12 and EZH2) were most efficiently precipitated and co-precipitated from fractions 18-26, indicating that PHF13 interacts with these proteins in this fraction pool (18-26) and supports the possibility of a common molecular complex containing RNA PolII, PRC2, PHF13, H3K4me3 and H3K27me3. Similarly, PHF13 co-precipitated with these proteins in the high molecular weight pool (13-17) albeit with substantially less PRC2. Finally, PHF13 co-precipitated only with RNA PolII S2P and S7P and with substantially reduced PRC2 in fractions (27-33). In this lower molecular weight pool RNA PolII S5P and histones were not detected. It is reported that PRC2 specifically interacts with RNA PolII S5P (Brooks et al., 2012), arguing that PHF13’s forms independent interactions with RNA PolII S2P, S7P and PRC2 in this lower molecular weight pool. Together, this argues that PHF13 forms different RNA PolII containing complexes.

The mass spectrometry analysis has proven to be the most difficult point in this revision, due to technical challenges. We have repeated the column chromatography in E14 ES cells and can show co-elution of PHF13 with PRC2 (EZH2 and SUZ12) in a subfraction of the eluted fractions. Furthermore, from the pooled PHF13 and PRC2 overlapping subfractions we are able to co-immunoprecipitate PHF13 and PRC2 together, supporting that they are in a complex together. Unfortunately, we have failed to visualize this interaction by mass spectrometry from PHF13 immunoprecipitated column chromatography pooled fractions. However, we did manage to see these interactions by mass spectrometry from classical PHF13 immunoprecipitation in E14 mESCs nuclease digested chromatin lysates (Figure 4).

We had two major problems contributing to these failures. One problem is that we have had recurrent PEG contamination from an unknown source, causing ion suppression and no peptides being visualized by MS. We were able to partially rescue this problem, by either running our samples first through a 4-15% gradient gel, or by using a detergent removal resin from Pierce prior to loading the samples on the MS. Under these conditions we successfully identified PHF13 interactions by MS (from classical immunoprecipitation and not column sorted material), however such cleaning methods also always results in a substantial loss of material. Since we were performing MS on endogenous PHF13 IPs, which is already not a very complex sample, we lost a substantial amount of interacting proteins/peptides, rendering this approach inappropriate for detection of PHF13 interacting proteins from column sorted and pooled PHF13 IPs, which only further reduces the complexity of the sample. Nevertheless, amongst the PHF13 interactions that we did identify from MS runs of PHF13 classical IPs were RNA polymerase II complex members, Polr2a, Polr2b, Polr2C and Polr2h as well as Suz12. Furthermore, we could confirm these interactions with SUZ12 and RNA polymerase II by reciprocal co-IP experiments. Additional interactions were also identified that have been previously observed and published such as ATM and TRIM28 (Mund et al., 2012) and others which we have not yet validated due to time restrictions.

The second major problem that we have had is the ability to see PHF13 in the MS results. We know from our IP/WB experiments that PHF13 was precipitated very efficiently and still we had difficulty to see this in our MS results. Therefore, we analyzed the ability of PHF13 to be proteolytically cleaved by trypsin, using in silico digestion and compare this with previously identified peptides from MS studies using the Protein Atlas website. It turns out that PHF13 is extremely inefficiently cleaved by trypsin in contrast to SUZ12 for example. In silico digestion in Protein Atlas returns the result that majority of the protein is unlikely to be observed by MS (https://db.systemsbiology.net/sbeams/cgi/PeptideAtlas/GetProtein?atlas_build_id=446&protein_name=Q8K2W6&action=QUERY) and that to date only 3 peptides have been reported in their data base for PHF13, whereas SUZ12 for example gives thousands of peptides from various studies with 34 unique peptides in total (https://db.systemsbiology.net/sbeams/cgi/PeptideAtlas/GetProtein?atlas_build_id=446&protein_name=ENSMUSP00000017692&action=QUERY). Consistently the study by Nikolov et al., 2011, which we have referenced in the manuscript, could only show two PHF13 peptides when allowing for mis-cleavages in their MS studies. When we allow for mis-cleavages in our MS runs we can detect one unique peptide for PHF13 in our IPs, but not in the controls. While this has been a challenging point to address, I am nevertheless appreciative of this information as it explains why PHF13 has not been previously seen in PRC2 mass spec analysis, and because we have identified PHF13 as a novel binding partner of RNA polymerase II.

2) You need to examine if there is any 'free' PHF13 in sESCs and interpret the ChIPseq data accordingly? Without this knowledge, it is very difficult to understand why there are so few overlapping ChIP-seq binding sites between PHF13 and SUZ12 or between PHF13 and EZH2.

The mass spectrometry analysis of PHF13 immunoprecipitations in combination with the column chromatography elution profiles and immunoprecipitations from pooled column fractions revealed that RNA PolII is a major interaction partner of PHF13 (Figure 4). Furthermore, since RNA PolII S5P forms both active and paused complexes, either with or without PRC2, where the former is a minority of RNA PolII S5P complexes, we interpret these data together to indicate that PHF13 can associate with both active and paused RNA PolII S5P complexes. Consistent with this interpretation genome wide overlap analysis of PHF13 with RNA PolII S2P, S5P, S7P and PRC2 (SUZ12 and EZH2), supports that PHF13 localizes to both active and PRC2 repressed promoters (Figure 5).

Furthermore, PHF13 showed a substantial overlap with promoter affiliated RNA PolII but not with the elongating form of RNA PolII (RNA PolII S2), as is expected due to PHF13’s affiliation with H3K4me2/3 and not with H3K36me3. Finally, PHF13 showed the strongest overlap with RNA PolII S5P, which occurred in both active and repressed promoter clusters. The larger cluster was enriched for H3K4me2/3, DHS, all forms of RNA PolII and CpG density and was depleted of H3K27me3 and PRC2, indicating active promoters whereas the smaller cluster was enriched for H3K4me2/3, H3K27me3, PRC2, RNA PolII S5 and CpG density and depleted of hypophosphorylated RNA PolII, RNA PolII S2P and RNA PolII S7P, reminiscent of bivalent promoters. Together, we interpret these observations to indicate that PHF13 co-localizes with promoter associated RNA PolII S5P at H3K4me2/3 demarcated active and repressed bivalent promoters. Since bivalent promoters only represent a small portion of H3K4me3, H3K27me3 and PRC2 binding sites genome wide and since they similarly represent only a small fraction of RNA PolII S5P binding sites, we find the approximate 10% representation in our data not inconclusive with this interpretation. Our data indicate that PHF13 localizes only to H3K4me2/3 containing promoters and not to H3K27me3 only promoters or to gene bodies where a substantial fraction of H3K27me3 and PRC2 peaks reside. Therefore, due to the reviewers’ suggestions that we should perform MS, which identified RNA PolII as an interaction partner that led us to analyze this relationship genome-wide with PHF13, we feel that we can now better explain this modest but significant overlap with PRC2 at PolII S5P, H3K4me2/3, H3K27me3 containing bivalent promoters.

3) You need to state what type of RNA was analyzed by RNA-seq. If it was total mRNA, then you need to tone down the interpretations on transcriptional regulation, because the state of the field now is to look at nascent RNA.

We have analyzed total RNA and not total mRNA in our RNAseq experiments. We used ribo-depletion to remove abundant rRNAs, tRNAs etc. and then sequenced the remaining RNA. This is described in the Materials and methods section. In addition to this, we now provide compelling evidence that PHF13 interacts with RNA polymerase II at H3K4me2/3 containing promoters (Figure 4 and Figure 5). Furthermore, PHF13 regulated genes showed enrichment in transcription, cell cycle, chromatin organization and development functional annotations (Figure 6), all of which with the exception of transcription have been previously reported as PHF13 related functions (Kinkley et al., 2009, Bordlein et al., 2011 and Mund et al., 2012). Together these observations argue that PHF13 interaction with RNA polymerase II at H3K4me2/3 containing promoters, impacts gene expression. Further in line with this interpretation, we show that PHF13 depletion reduces not only the binding of SUZ12, RNA PolII S2P and S5P for PHF13, but that it also reduces the binding of SUZ12 and RNA PolII S5P for H3K4me3 and H3K27me3 and with one other, indicating that it is required for complex stability and/or affiliation of these complexes with H3K4me3 (Figure 7). Together we hope that these new insights convince the reviewers that PHF13 plays a role in co-transcriptional regulation.

4) Validation of the ChIP-seq results is required by ChIP-qRT PCR for example.

To address the concern that “Experimentally, there is no validation of the results of a single PHF13 ChIP-seq by alternative methods”, we validated several of the PHF13 ChIP sequencing targets that were additionally observed as PHF13 RNA sequencing targets (Figure 6—figure supplement 1) by ChIP qPCR (Figure 5—figure supplement 1). All of the regions analyzed showed a greater than 5-fold enrichment over an IgG control, in contrast to two selected control regions (Figure 5—figure supplement 1).

5) Demonstration that similar RNA changes were observed with more than one siRNA is necessary to prove these were not off target effects. This could also be done by qRT PCR.

To address the concern that the RNA-seq was done using only one shRNA, we confirmed several of the genes that were both up and down regulated in our RNA sequencing results by qPCR with a second independent and inducible shRNA (Figure 6—figure supplement 1) as well as reproduced the RNA sequencing results of PHF13 shRNA Sh3. The sequences of both shRNA’s are reported in the Materials and methods. What was nice about this confirmation is that the original stable viral shRNA cell lines, took 12 days to establish due the selection process of the transduced cells. This indicates that gene expression changes that we are observing are potentially late occurring consequences due to PHF13 depletion. In the confirmation experiment we used a Doxycycline inducible PHF13 shRNA cell line which allowed us to look at the regulation of several up and down regulated genes at both an earlier time point (48h) and at a similar time point to the original experiment (11d) after PHF13 depletion. Interesting the vast majority of genes already showed expression changes by 48h and all showed the expected changes by 11d. However, there were a few genes that did not change at the early time point that did show the corresponding change by 11d, showing a kinetics or dynamics to gene regulation after PHF13 at certain genes that cannot be observed by evaluating single time points. Together these results demonstrate that several PHF13 target genes were reproducibly altered in expression by an independent PHF13 shRNA, arguing that the gene expression changes are not due to off target effects. Furthermore, the results indicate which could not be determined in the original experiments due to technical drawbacks of selection, that the vast majority of these genes are changed already by 48h post PHF13 induced depletion.

*6) Determine whether the interaction between the PHF13 PHD finger and histone H3K4 methylation is required for PHF13's function.*

In an effort to demonstrate that PHF13 localization at H3K4me3 is required for its proposed function in transcriptional regulation, we employed an inducible PHF13 shRNA mESC cell line. We then evaluated the ability of PHF13 associated complexes, PRC2 (SUZ12) and RNA PolII (S2P and S5P) to interact with PHF13, H3K4me3, H3K27me3 and with each other in wild type or PHF13 depleted lysates (Figure 7). Our results demonstrate that PHF13 depletion not only leads to its reduced precipitation (PHF13) and co-precipitation by SUZ12, RNA PolII S2P and RNA PolII S5P, but that it also disrupts the interaction between both SUZ12 and RNA PolII S5P with H3K4me3 and H3K27me3 and with each other. These findings indicate that PHF13 is important for complex stability and or localization giving mechanistic insight into PHF13’s ability to modulate transcriptional activity. These findings in part address the concerns of Reviewer 1, providing mechanism and linking PHF13’s functions to its in vitro biochemical properties. Furthermore, these findings implicate PHF13 in PRC2 recruitment and potentially in the establishment of bivalent chromatin as questioned by Reviewer 1 “whether PHF13 is required for PRC2 recruitment to chromatin and the establishment of bivalent domains”. We show that PHF13 interacts with PRC2 and H3K4me2/3 (Figure 2–Figure 4), that it shows a genome wide co-localization at a subset of PRC2 and H3K27me3 targets that resemble bivalent promoters according to their molecular features (Figure 5) and that its depletion leads to a decreased interaction of SUZ12 with H3K4me3 and H3K27me3 (Figure 7), as well as upregulation of genes that are normally enriched for PRC2 (Figure 7). Furthermore, PHF13 is found to co-localize and interact with active promoters features (H3K4me2/3, RNA PolII S5P and S7P and CpG rich DNA), which are down regulated following PHF13 depletion and RNA PolII S5P shows a decreased association with H3K4me3 after PHF13 depletion. These findings provide mechanistic insight into how PHF13 plays as role in active transcription, as was questioned by Reviewer 1 “How does PHF13 activate gene expression”, i.e., by bridging activating PolII S5P complexes to H3K4me3 promoters. Together these findings argue that PHF13 is a transcriptional co-regulator of both active and repressed chromatin and that “the epigenetic consequences upon PHF13 depletion” as questioned by Reviewer 1, are the altered chromatin localization of PolII S5 containing transcriptional activating and repressing complexes and altered gene expression. Supporting this statement the PHF13 ChIP sequencing targets and the genes that are altered after PHF13 depletion are enriched in transcription, development, cell cycle and chromatin structure, all of which are described PHF13 functional roles if you include the results from this manuscript and previous publications (Kinkley et al., 2009, Bordlein et al., 2011 and Mund et al., 2012).

While we did not directly address the question of “the role of the DNA interaction of PHF13 and if there is a synergy between DNA binding and histone binding of PHF13 in cells” posed by Reviewer 1, our data indicate 1) that PHF13 can interact directly with both DNA and H3K4me2/3 via independent domains, 2) that PHF13 ChIP seq targets are enriched for CpG rich sequences, and 3) that PHF13 co-localizes at both active and repressed chromatin regions which are enriched for H3K4me2/3 and CpG rich DNA overlapping at PHF13 peaks. These features are common to both active and bivalent promoters and indicate that they could and likely do cooperate in PHF13 recruitment as they co-occur at all PHF13 peaks (Figure 5). It is important to note that Figure 5 is not in contradicting of this statement as PHF13 only shows a 55% overlap with annotated CpG islands. This is due to the fact that not all CpG rich DNA sequences are classified as CpG islands, which has other criteria than simply the CpG content. Furthermore, while it is only theoretical, the ability of PHF13 to simultaneously and directly affiliate with DNA and histones would increase the avidity of PHF13 for chromatin making its binding constant for H3K4me3/CpG rich promoters in the physiological range and is a mechanism “multivalent binding” used by many chromatin readers and effectors.